# FORGET THE DATA AND FINE-TUNING! JUST FOLD THE NETWORK TO COMPRESS

Dong Wang$^{\diamond,*}$, Haris Šikić$^{\diamond,*}$, Lothar Thiele$^\dagger$, Olga Saukh$^{\diamond,\S}$

$^\diamond$Graz University of Technology, Austria
$^\S$Complexity Science Hub Vienna, Austria
$^\dagger$ETH Zurich, Switzerland
{dong.wang@, haris.sikic@student., saukh@}tugraz.at
thiele@tik.ee.ethz.ch

## ABSTRACT

We introduce *model folding*, a novel data-free model compression technique that merges structurally similar neurons across layers, significantly reducing the model size without the need for fine-tuning or access to training data. Unlike existing methods, model folding preserves data statistics during compression by leveraging $k$-means clustering, and using novel data-free techniques to prevent variance collapse or explosion. Our theoretical framework and experiments across standard benchmarks, including ResNet18 and LLaMA-7B, demonstrate that model folding achieves comparable performance to data-driven compression techniques and outperforms recently proposed data-free methods, especially at high sparsity levels. This approach is particularly effective for compressing large-scale models, making it suitable for deployment in resource-constrained environments.

## 1 INTRODUCTION

Deep neural networks (DNNs) have emerged as a fundamental technology, driving progress across a multitude of applications from natural language processing to computer vision. However, the deployment of these models in real-world settings is often constrained by the computational and memory resources available, particularly on edge devices like smartphones and embedded systems (Wan et al., 2020; Kumar et al., 2017; Chen et al., 2020). This limitation poses a significant challenge, as the growing complexity and size of SOTA models demand increasingly substantial resources (Bommasani et al., 2021; Chang et al., 2024; Rombach et al., 2022).

Conventional model compression techniques, such as pruning (Han et al., 2015; LeCun et al., 1989; Li et al., 2016b; Hassibi et al., 1993) and quantization (Gupta et al., 2015; Zhou et al., 2017; Li et al., 2016a), have been developed to mitigate this issue by reducing the model size and computational requirements. These methods usually remove redundant or less critical parameters from the model, thereby reducing the overall size and computational load. For example, pruning eliminates weights that contribute minimally to the model's output (Han et al., 2015; Li et al., 2016b). Quantization reduces the precision of the weights and activations (Gupta et al., 2015), which decreases memory usage and speeds up inference (Zhou et al., 2017). Despite their effectiveness, these approaches often introduce a degradation in model performance, necessitating a phase of fine-tuning to maintain the internal data statistics within the model (Jordan et al., 2022) and restore the original accuracy levels (Frankle & Carbin, 2018; Hassibi et al., 1993; Frantar & Alistarh, 2022). This requirement can be a significant drawback in scenarios where access to the original training data is limited.

Recent methods have sought to circumvent the need for extensive retraining or fine-tuning by exploring alternatives to traditional approaches. Instead, several recent strategies build on model merging techniques (Entezari et al., 2022; Ainsworth et al., 2023; Jordan et al., 2022) and achieve (multi-)model compression by fusing similar computational units. For example, ZipIt! (Stoica et al., 2024) merges two models of the same architecture by combining similar features both within and

---

$^*$Equal contribution.

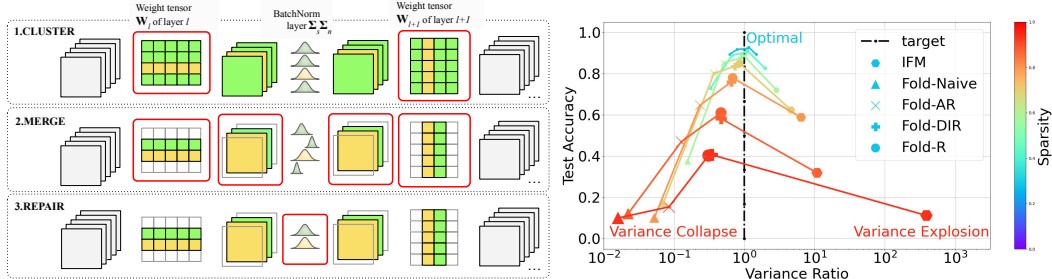

Figure 1: **Model compression and repair of data statistics. Left:** Model folding pipeline is applied layer-wise, consisting of three phases: weight tensor clustering and merging, and data statistics repair. **Right:** To maintain accuracy, the data variances of compressed and uncompressed models must align (*i.e.*, the variance ratio must be close to 1), as variance collapse or explosion leads to suboptimal performance. Our data-free and fine-tuning-free model folding methods (Fold-AR and Fold-DIR) achieve performance comparable to data-driven statistics repair (Fold-R), while outperforming naive statistics repair (Fold-naive) and the recently proposed IFM (Chen et al., 2023). All methods were evaluated on a public ResNet18 checkpoint trained on CIFAR10. Lines connect the performance of different methods at the same weight sparsity level, applied uniformly across all layers. Variance ratio refers to the activation outputs in the last layer. A precise definition and analysis are in Sec. 3.

across models. They provide both theoretical and empirical evidence suggesting that features within the same model are more similar than those between models trained on different tasks. This method avoids the need for retraining the compressed model but requires training data to match features based on the similarity of their activations. Similarly, Yamada et al. (2023) examine various model merging techniques and conclude that merged models require a dataset—such as a coreset—for effective merging and to achieve high accuracy. This data is essential for adjusting internal data statistics that are disrupted by weight fusion, such as updating the running mean and variance in BatchNorm layers (Ioffe & Szegedy, 2015). The process involves a simple forward pass through the model and is a well-established method to adapt models in low-resource environments (Leitner et al., 2023).

In contrast, IFM (Chen et al., 2023) offers a fully data-free and fine-tuning-free approach, utilizing weight matching (Ainsworth et al., 2023) to iteratively merge similar hidden units, similar to Stoica et al. (2024). However, despite a heuristic for preserving data statistics, we demonstrate that IFM fails to maintain performance across standard architectures and for high sparsity. Other data-free approaches, such as (Yin et al., 2020), generate synthetic images directly from the uncompressed model for fine-tuning to restore pruned model accuracy. More related work is covered in Appendix N.

This paper presents a model compression technique, *model folding*, that exploits weight similarity through three phases: neuron clustering, merging, and data statistics repair, summarized in Fig. 1 (left). We demonstrate that $k$-means clustering provides a theoretically optimal and data-free method for merging weights. Building on Jordan et al. (2022), which addresses variance collapse using REPAIR with training data, we introduce two data-free alternatives: Fold-AR (folding with approximate REPAIR) and Fold-DIR (folding with Deep Inversion-based REPAIR). Fold-AR estimates mean correlations within clusters assuming independent inputs, while Fold-DIR uses Deep Inversion (Yin et al., 2020) to synthesize a single batch of images for updating BatchNorm statistics via a forward pass. Both methods maintain data statistics and prevent variance collapse or explosion to avoid suboptimal compression performance, with Fold-AR standing out as a more resource-efficient option while still significantly surpassing existing methods. Fig. 1 (right) shows that the highest accuracy at any target sparsity is achieved when the mean variance ratio over the last layer between the compressed and uncompressed models stays close to one. Our contributions are:

- We introduce *model folding*, a novel model compression technique that merges structurally similar neurons within the same network to achieve compression. We provide both theoretical and empirical evidence demonstrating that $k$-means clustering is an optimal and effective method for fusing model weights in a data-free manner.

- To enable data-free model compression, we adapt the REPAIR framework proposed by Jordan et al. (2022) to address variance collapse of data statistics within a model after

layer compression. We introduce *data-free* and *fine-tuning-free* versions of REPAIR, that effectively maintain model statistics and achieve high performance.

- We demonstrate that model folding surpasses the performance of SOTA model compression methods which do not use data or fine-tune the pruned model, including recently proposed IFM (Chen et al., 2023), and INN (Solodskikh et al., 2023), in particularly at higher levels of sparsity and when applied to more complex datasets.

- We use model folding on LLaMA-7B without utilizing data or post-tuning and achieve comparable results to methods that require data and fine-tuning.

## 2 PRELIMINARIES

Our work is inspired by recent advances in two key areas: neuron alignment algorithms for fusing model pairs in weight space, and data-driven methods for recovering from variance collapse in fused models. Below, we summarize the relevant results from the literature.

**Neuron alignment algorithms**. Model merging involves combining the parameters of multiple trained models into a single model, with a key challenge being the alignment of neurons across these models, particularly when they are trained on different datasets or tasks. Neuron alignment methods can be classified based on their dependency on the input data. Methods like the Straight Through Estimator (STE) (Ainsworth et al., 2023), Optimal Transport (OT) (Singh & Jaggi, 2020) and correlation-based activation matching (Li et al., 2015) require data for effective merging. In contrast, weight matching (Yamada et al., 2023; Ainsworth et al., 2023) is a data-free method, making it efficient in scenarios when training data is not available. In weight matching, neurons are aligned by minimizing the $L_2$ distance between the weight vectors of neurons across models. Given two models with weight matrices $\mathbf{W}_A$ and $\mathbf{W}_B$, the goal is to find a permutation $\mathbf{P}$ of the weights in $\mathbf{W}_B$ that minimizes the distance:

$$\min_{\mathbf{P}} \|\mathbf{W}_A - \mathbf{P}\mathbf{W}_B\|_2^2,$$

where $\mathbf{P}\mathbf{W}_B$ denotes the weight matrix $\mathbf{W}_B$ after applying the permutation $\mathbf{P}$ to align it with $\mathbf{W}_A$. Once the optimal permutation is found, the models are merged by averaging the aligned weights:

$$\mathbf{W}_{\text{merged}} = \frac{1}{2}\left(\mathbf{W}_A + \mathbf{P}^*\mathbf{W}_B\right),$$

where $\mathbf{P}^*$ is the permutation that minimizes the $L_2$ distance. Weight matching solves an instance of the linear sum assignment problem (LSAP), usually solved by Hungarian algorithm (Kuhn, 1955) as done in (Jordan et al., 2022; Ainsworth et al., 2023), to layer-wise align weight vectors. Unlike merging different models, aligning neurons within a single model requires an acyclic matching graph, a challenge not addressed by LSAP, which assumes disjoint task and worker sets. To overcome the challenge Chen et al. (2023) and He et al. (2018) apply iterative approach greedily merging a pair of the most similar neurons in each iteration. This work extends weight matching to align *clusters* of similar neurons within the same model, remaining data-free. Appendix C provides more details on the relationship between weight matching and model folding. We show that IFM is inferior to clustering utilized by model folding as described in the next section.

**Variance collapse and REPAIR**. When interpolating between independently trained, neuron-aligned networks, (Jordan et al., 2022) observed a phenomenon they termed *variance collapse*. This occurs when the variance of hidden unit activations in the interpolated network significantly diminishes compared to the original networks, leading to a steep drop in performance. To solve this issue, Jordan et al. (2022) introduced the REPAIR method (Renormalizing Permuted Activations for Interpolation Repair) which uses input data to recompute the internal data statistics.

REPAIR works by rescaling the preactivations of the interpolated network to restore the statistical properties of the original networks. Specifically, it adjusts the mean and variance of the activations in each layer of the interpolated network to match those of the corresponding layers in the original networks. This is done by computing affine transformation parameters—rescaling and shifting coefficients—for each neuron, ensuring that the mean and standard deviation of activations in the interpolated network are consistent with those in the original models. REPAIR effectively mitigates the variance collapse, enabling the interpolated network to maintain performance closer to that of

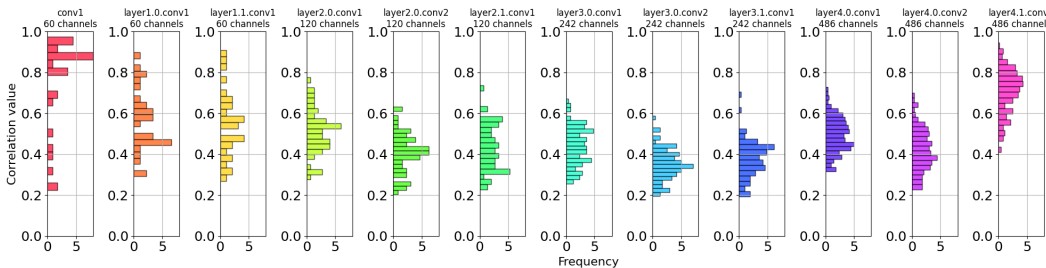

Figure 2: **Layer-wise correlation between matched channels in ResNet18 trained on CIFAR10**. For each layer, we use activation matching with $L_2$ distance measure to greedily pair similar neurons. Each subplot shows the correlation within all matched pairs.

the original models. This technique has become essential in recent work to preserve model accuracy after merging (Ainsworth et al., 2023; Yamada et al., 2023; Jolicoeur-Martineau et al., 2024). While REPAIR relies on input data to preserve the network's statistical properties, this paper proposes a data-free alternative.

## 3 MODEL FOLDING

In this section, we introduce *model folding*, a novel compression technique that reduces the computational complexity and size of neural networks by merging similar neurons in each layer without requiring training data. As illustrated in Fig. 1 (left), model folding processes the network layer by layer, involving filter clustering, merging, and correcting data statistics. Below, we present a theoretical analysis of our approach, supported by empirical results on ResNet18 using CIFAR10.

### 3.1 CHANNEL CLUSTERING

**Channel similarity**. Neural networks trained with stochastic gradient descent (SGD) tend to have many correlated hidden units, as illustrated in Fig. 2. Model folding exploits this observation, which is related to the implicit bias of SGD. As discussed in (Gunasekar et al., 2017), SGD exhibits a minimum norm bias, which can be viewed as a form of regularization when no explicit regularization is used. In contrast to $L_1$ regularization, which promotes sparsity, the minimum Euclidean norm solution ($L_2$ norm) penalizes large weights, encouraging smaller, more regular weights. This not only prevents overfitting but also results in smoother decision boundaries (Bishop, 2006). While the minimum norm solution does not directly enforce weight similarity, we empirically demonstrate in Appendix D that it leads to effective model compression when applying similarity-based methods. Recently published methods (Stoica et al., 2024; Chen et al., 2023) leverage the same observation.

**Folding as a clustering problem**. This work extends weight matching (Ainsworth et al., 2023), which minimizes the $L_2$ distance between weight vectors and operates without requiring training data. Instead of finding pairs of similar neurons by solving the linear sum assignment problem (LSAP) with a Hungarian algorithm (Kuhn, 1955) as done in (Jordan et al., 2022; Ainsworth et al., 2023), we achieve channel matching using $k$-means clustering. In the following, we justify this approach as it provides an optimal weight matrix approximation.

Given a neural network layer $l$ with a weight matrix $\mathbf{W}_l \in \mathbb{R}^{n \times m}$, we define the output of this layer as $\mathbf{y}_l = \sigma(\mathbf{W}_l \mathbf{x}_l)$, where $\mathbf{x}_l \in \mathbb{R}^m$ is the input vector to this layer, $\mathbf{y}_l \in \mathbb{R}^n$ is the output vector, and $\sigma(\cdot)$ is a non-linear activation function applied element-wise.

To reduce the number of outputs of layer $l$ we cluster (fold) rows of $\mathbf{W}_l$, i.e., $k$ cluster centroids are determined which serve as a prototype of the respective cluster of rows. All rows of a cluster are replaced by their cluster centroid. This can be formulated as

$$\mathbf{W}_l \approx \mathbf{UM},$$

where $\mathbf{M} \in \mathbb{R}^{k \times m}$ contains the $k < n$ cluster centroids and the cluster matrix $\mathbf{U} \in \{0, 1\}^{n \times k}$ determines the membership of a row: $u(i, j) = 1$ if the $i$-th row of $\mathbf{W}_l$ belongs to the $j$-th cluster, and $u(i, j) = 0$ otherwise.

As a measure of the approximation error when replacing the rows of $\mathbf{W}_l$ by $k < n$ prototypes, we use the Frobenius norm $\| \cdot \|_F^2$ of the difference between $\mathbf{W}_l$ and the low-rank factorization $\mathbf{UM}$:

$$J = \|\mathbf{W}_l - \mathbf{UM}\|_F^2 = \operatorname{tr}(\mathbf{W}_l \mathbf{W}_l^T) + \operatorname{tr}(\mathbf{UMM}^T \mathbf{U}^T) - 2\operatorname{tr}(\mathbf{UMW}_l^T).$$

We determine the optimal matrix of cluster centroids by setting the derivative of $J$ with respect to $\mathbf{M}$ to zero:

$$\mathbf{M} = (\mathbf{U}^T \mathbf{U})^{-1} \mathbf{U}^T \mathbf{W}_l.$$

As a result, we can write

$$\mathbf{W}_l \approx \mathbf{UM} = \mathbf{CW}_l \quad \text{with} \quad \mathbf{C} = \mathbf{U}(\mathbf{U}^T \mathbf{U})^{-1} \mathbf{U}^T.$$

As mentioned above, we use $k$-means clustering for folding as this minimizes $J$ by determining the optimal clustering matrix $U$ and the corresponding cluster centroids $M$, also see (Bauckhage, 2015).

**Interdependence between layers**. We will expand the above result to successive layers $l$ and $l + 1$. For simplicity of notation, we neglect the bias and get

$$\mathbf{y}_{l+1} = \sigma(\mathbf{W}_{l+1} \sigma(\mathbf{W}_l \mathbf{x}_l)).$$

Following the above notation, we describe the folding of activations by some clustering matrix $\mathbf{U}$ and $\mathbf{C} = \mathbf{U}(\mathbf{U}^T \mathbf{U})^{-1} \mathbf{U}^T$. It is shown in Appendix B that the corresponding approximation satisfies

$$\tilde{\mathbf{y}}_{l+1} = \sigma(\mathbf{W}_{l+1} \sigma((\mathbf{CW}_l) \mathbf{x}_l)) = \sigma((\mathbf{W}_{l+1} \mathbf{C}^T) \sigma((\mathbf{CW}_l) \mathbf{x}_l)).$$

Adding up the individual folding costs $J_{l+1} = \|\mathbf{W}_{l+1}^T - \mathbf{CW}_{l+1}^T\|_F^2$ and $J_l = \|\mathbf{W}_l - \mathbf{CW}_l\|_F^2$ yields the combined approximation error $J_{l,l+1} = J_{l+1} + J_l$ for folding layer $l$ which can be rewritten as

$$J_{l,l+1} = \|\mathbf{W}_{l,l+1} - \mathbf{CW}_{l,l+1}\|_F^2 \quad \text{with} \quad \mathbf{W}_{l,l+1} = \begin{bmatrix} \mathbf{W}_l \mid \mathbf{W}_{l+1}^T \end{bmatrix}.$$

If we perform $k$-means clustering on $\mathbf{W}_{l,l+1}$ and use the resulting clustering matrix $\mathbf{U}$ in $\mathbf{C} = \mathbf{U}(\mathbf{U}^T \mathbf{U})^{-1} \mathbf{U}^T$, then the combined approximation error $J_{l,l+1}$ is minimized. This approach accounts for the impact of compressing one layer on the next, leading to more efficient compression that balances the process and preserves learned representations while reducing model size. Our folding methods outperforms other methods experimentally, see Fig. 3 for a comparison to other clustering methods and Iterative Greedy (greedy) adopted in SOTA.

**Batch Normalization**. Now, let us consider batch normalization in layer $l$ represented by two diagonal matrices $\mathbf{\Sigma}_s$ (scaling) and $\mathbf{\Sigma}_n$ (normalization), again neglecting the bias to reduce notation. In this case, we get

$$\mathbf{y}_{l+1} = \sigma(\mathbf{W}_{l+1} \sigma(\mathbf{\Sigma}_s \mathbf{\Sigma}_n \mathbf{W}_l \mathbf{x}_l)).$$

The folding of layer $l$ can be distributed to the matrices $\mathbf{\Sigma}_s$, $\mathbf{\Sigma}_n$, and $\mathbf{W}_l$ in various ways, depending on the chosen correction of the variance, see Sec. 3.2. For example, one can cluster each matrix separately, leading to

$$\tilde{\mathbf{y}}_{l+1} = \sigma((\mathbf{W}_{l+1} \mathbf{C}^T) \sigma((\mathbf{C\Sigma}_s)(\mathbf{C\Sigma}_n)(\mathbf{CW}_l) \mathbf{x}_l)).$$

Adding up the individual folding costs $J_{l+1}$, $J_s$, $J_n$, and $J_l$ for each of the matrices $\mathbf{W}_{l+1}$, $\mathbf{\Sigma}_s$, $\mathbf{\Sigma}_n$ and $\mathbf{W}_l$, respectively, yields the total approximation error $J_{\text{tot}} = J_{l+1} + J_s + J_n + J_l$ for folding layer $l$

$$J_{\text{tot}} = \|\mathbf{W}_{\text{tot}} - \mathbf{CW}_{\text{tot}}\|_F^2 \quad \text{with} \quad \mathbf{W}_{\text{tot}} = \begin{bmatrix} \mathbf{W}_{l+1}^T \mid \mathbf{W}_l \mid \operatorname{diag}(\mathbf{\Sigma}_s) \mid \operatorname{diag}(\mathbf{\Sigma}_n) \end{bmatrix}$$

If we perform $k$-means clustering on $\mathbf{W}_{\text{tot}}$ then the total approximation error $J_{\text{tot}}$ is minimized. This approach is used in the Deep Inversion (DI) REPAIR, see next section.

Instead, if we decompose the folding of layer $l$ according to

$$\tilde{\mathbf{y}}_{l+1} = \sigma((\mathbf{W}_{l+1} \mathbf{C}^T) \sigma((\mathbf{C\Sigma}_s)(\mathbf{C\Sigma}_n \mathbf{W}_l) \mathbf{x}_l)).$$

then the individual folding costs of $\mathbf{W}_{l+1}$, $\mathbf{\Sigma}_s$ and the normalized weight matrix $\mathbf{\Sigma}_n \mathbf{W}_l$ add up to

$$J_{\text{tot}} = \|\mathbf{W}_{\text{tot}} - \mathbf{CW}_{\text{tot}}\|_F^2 \quad \text{with} \quad \mathbf{W}_{\text{tot}} = \begin{bmatrix} \mathbf{\Sigma}_n \mathbf{W}_l \mid \operatorname{diag}(\mathbf{\Sigma}_s) \mid \mathbf{W}_{l+1}^T \end{bmatrix}.$$

Again, if we perform $k$-means clustering on this combined matrix $\mathbf{W}_{\text{tot}}$ then the corresponding total approximation error $J_{\text{tot}}$ is minimized. This approach is used in the approximate REPAIR, see Sec. 3.2. For completeness, we present in Appendix F how we handle residual connections.

**Merging similar channels in each cluster**. To fuse similar channels, various approaches have been proposed in the literature, such as fusing weights for multitasking, which involves Hessian calculations (He et al., 2018), or by combining the matched weights into a single channel (Chen et al., 2023). (Matena & Raffel, 2022) introduces Fisher-weighted averaging based on the Laplace approximation for merging weights, while (Jin et al., 2023) suggests computing a regression mean, which is both computationally efficient and scalable for merging multiple models. In our approach, we use above formulation of the optimization problem as $k$-means clustering and use a simple mean to compute the cluster centroids.

### 3.2 MAINTAINING DATA STATISTICS IN A COMPRESSED MODEL

**Variance collapse and variance overshooting**. We use the conceptual framework in (Jordan et al., 2022) to analyze the performance of model compression methods. We use the following definition.

**Definition 1** (Variance ratio). *Consider a neural network $f(\mathbf{x}, \boldsymbol{\Theta})$ with layer activations $\{\mathbf{x}_l\}_1^L$ and its compressed version $\tilde{f}(\mathbf{x}, \boldsymbol{\Theta})$ with activations $\{\tilde{\mathbf{x}}_l\}_1^L$. The* variance ratio *of the l-th layer is:*

$$\mu\left[\frac{\mathrm{Var}(\tilde{\mathbf{x}}_l)}{\mathrm{Var}(\mathbf{x}_l)}\right] = \frac{1}{|\mathbf{x}_l|}\sum_{k=1}^{|\mathbf{x}_l|}\frac{\mathrm{Var}(\tilde{\mathbf{x}}_{l,k})}{\mathrm{Var}(\mathbf{x}_{l,k})}.$$

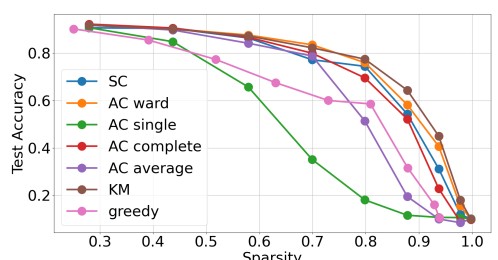

We observe not only variance collapse but also variance overshooting phenomena. Specifically, when data statistics are not accurately corrected after channel merging, as in IFM, variance overshooting can occur, leading to network performance decline. Fig. 4 shows layerwise variance ratio between the compressed and uncompressed networks. Staying close to 1 is essential to mitigate both phenomena. This highlights the critical need for precise statistical corrections during model merging.

Figure 3: $k$**-means (KM) outperforms other clustering methods**: Spectral Clustering (SC), Agglomerative Clustering (AC) with different linkage criteria and Iterative Greedy (greedy) used to compress ResNet18 trained on CIFAR10. Data-based REPAIR was used to restore data statistics after clustering for all methods.

**Fold-AR: Folding with approximate REPAIR**. In the context of model compression, particularly when using folding as a clustering method, it is crucial to ensure that the compressed model maintains accurate data statistics. This is especially important for layers involving operations like BatchNorm, where maintaining the correct statistical properties of activations is vital for model performance (Jordan et al., 2022; Yamada et al., 2023).

In the following explanation of the data-free approximate REPAIR, we neglect biases for ease of notation. Following the previous section, we consider folding of the normalized weight matrix with

$$\mathbf{z}_l = \mathbf{C}\boldsymbol{\Sigma}_n\mathbf{W}_l\mathbf{x}_l$$

using the post-activation output $\mathbf{x}_l$ of the previous layer and the input $\mathbf{z}_l$ to the scaling matrix $\boldsymbol{\Sigma}_s$. A cluster $c$ is defined by the column of the clustering matrix $U$, i.e., all values $z_l(i)$ with $u(i, c) = 1$ belong to cluster $c$. Moreover, by definition of $\mathbf{C}$, all values $z_l(i)$ belonging to a single cluster $c$ equal the centroid $\hat{z}_l(c)$ of the cluster, i.e., the average of all values $\boldsymbol{\Sigma}_n\mathbf{W}_l\mathbf{x}_l$ belonging to this cluster. More formally,

$$\forall u(i, c) == 1 \ : \ z_l(i) = \hat{z}_l(c)$$

$$\forall 1 \leq c \leq k \ : \ \hat{z}_l(c) = \frac{1}{N_c}\sum_{i \in I_c}\tilde{x}_l(i),$$

where $I_c = \{i : u(i, c) = 1\}$ denotes the indices of all values belonging to cluster $c$, $N_c = |I_c|$ denotes the number of values in the cluster, and $\tilde{\mathbf{x}}_l = \boldsymbol{\Sigma}_n\mathbf{W}_l\mathbf{x}_l$. The batch normalization using $\boldsymbol{\Sigma}_n$ ensures that the variances of all $\tilde{x}_l(i)$ equal 1. The averaging over all $\tilde{x}_l(i)$ belonging to a single cluster destroys this property and leads to the observed variance collapse. We will describe various methods to compensate this loss in variance, at first the data-free approximate REPAIR (Fold-AR).

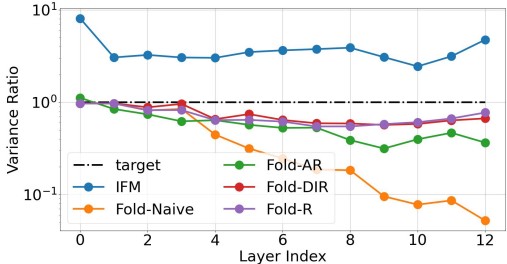 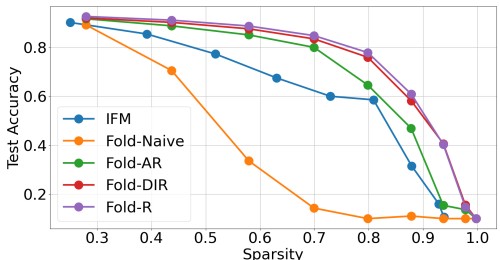

Figure 4: **Variance collapse and overshooting** on ResNet18 with CIFAR10. The goal is to align the layer-wise variance in the compressed network to that of the uncompressed model. Naive averaging of statistics (Fold-Naive) leads to variance collapse (Jordan et al., 2022), while IFM overshoots. Fold-AR and Fold-DIR closely match the performance of the data-driven REPAIR (Fold-R). Layer-wise sparsity is 0.5.

Figure 5: **Data-free folding methods** with approximate REPAIR (Fold-AR) and Deep Inversion (Yin et al., 2020) (Fold-DIR) and on ResNet18 with CIFAR10 at various weight sparsity levels, uniformly distributed across layers. Fold-DIR performs similarly to the data-based REPAIR (Fold-R). Both Fold-AR and Fold-DIR surpass IFM (Chen et al., 2023) by a significant margin.

The variance of the cluster centroid $\hat{z}_l(c)$ of cluster $c$ is given by

$$\mathrm{Var}(\hat{z}_l(c)) = \frac{1}{N_c^2}\left[\sum_{i \in I_c}\mathrm{Var}(\tilde{x}_l(i)) + \sum_{i,j \in I_c; i \neq j}\mathrm{Cov}(\tilde{x}_l(i), \tilde{x}_l(j))\right],$$

which further simplifies to $\mathrm{Var}(\hat{z}_l(c)) = \frac{1}{N_c^2}\left[N_c + (N_c^2 - N_c)E[c]\right]$, where $E[c]$ is the mean correlation within the cluster. To prevent variance collapse, we aim for $\mathrm{Var}(\hat{z}_l(c)) = 1$, which would occur if $E[c] = 1$, meaning all channels in the cluster are fully correlated. However, as $E[c] < 1$ typically, we multiply each cluster centroid by a scaling parameter assuming an average cluster correlation $E[c]$

$$\hat{z}_l(c) \leftarrow \hat{z}_l(c)\frac{N_c}{\sqrt{N_c + (N_c^2 - N_c)E[c]}}.$$

Suppose now that the covariance matrix $\boldsymbol{\Sigma}_{x_l}$ of the output $\mathbf{x}_l$ of the previous layer is available and that we define the normalized weight matrix $\tilde{\mathbf{W}}_l = \boldsymbol{\Sigma}_n \mathbf{W}_l$ with rows $\tilde{\mathbf{w}}_l(i)$. Then the correlation $E[c]$ can be computed as:

$$E[c] = \frac{1}{N_c^2 - N_c}\sum_{i,j \in I_c; i \neq j}\frac{\tilde{\mathbf{w}}_l(i)\boldsymbol{\Sigma}_{x_l}\tilde{\mathbf{w}}_l^T(j)}{\sqrt{(\tilde{\mathbf{w}}_l(i)\boldsymbol{\Sigma}_{x_l}\tilde{\mathbf{w}}_l^T(i))(\tilde{\mathbf{w}}_l(j)\boldsymbol{\Sigma}_{x_l}\tilde{\mathbf{w}}_l^T(j))}}.$$

In the absence of data, $E[c]$ can be estimated by assuming that the output values $\mathbf{x}_l$ of the previous layer are uncorrelated. As the individual variances of $\tilde{x}_l(i)$ equal 1 we obtain

$$E[c] = \frac{1}{N_c^2 - N_c}\sum_{i,j \in I_c; i \neq j}\frac{\tilde{\mathbf{w}}_l(i)\tilde{\mathbf{w}}_l^T(j)}{\sqrt{(\tilde{\mathbf{w}}_l(i)\tilde{\mathbf{w}}_l^T(i))(\tilde{\mathbf{w}}_l(j)\tilde{\mathbf{w}}_l^T(j))}}.$$

We term this approach to maintain the data statistics within the model *folding with approximate REPAIR* (Fold-AR). This approach helps to ensure that the statistical properties of the data are preserved even after model compression, maintaining the performance of the network while reducing its size. Fig. 5 shows how the performance of Fold-AR compares to the data-driven REPAIR (Fold-R) and surpasses the SOTA data-free methods.

**Fold-DIR: Correcting data statistics with deep inversion**. Deep Inversion (DI) (Yin et al., 2020) is a technique that synthesizes realistic images directly from a pre-trained neural network without requiring access to the original data. The process involves inverting the model by optimizing random noise to produce class-conditional images that match the statistics of the data the model was trained on (Mordvintsev et al., 2015). DI leverages the BatchNorm layers within the network, which store

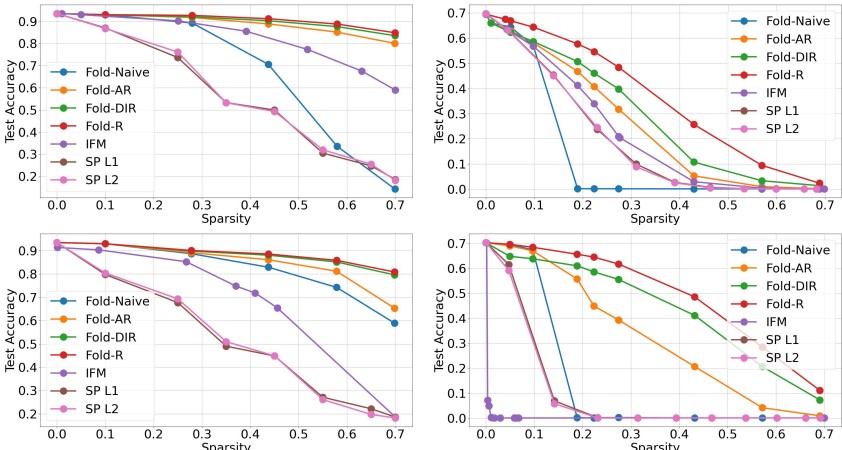

Figure 6: **Comparison with IFM (Chen et al., 2023) and structured magnitude pruning (Cai et al., 2020; Yin et al., 2022).** Model folding, when tested on ResNet18 (**top row**) and VGG11-BN (**bottom row**) trained on CIFAR10 (**left column**) and ImageNet (**right column**), outperforms IFM with higher sparsity and increasing dataset difficulty.

the running mean and variance of activations during training. By using these stored statistics as a regularization term in

$$\mathcal{R}(\hat{\mathbf{x}}) = \mathcal{L}_{class}(\hat{\mathbf{x}}, t) + \sum_l \|\mu(\hat{\mathbf{x}}_l) - \mu(\mathbf{x}_l)\|_2^2 + \sum_l \|\text{Var}(\hat{\mathbf{x}}_l) - \text{Var}(\mathbf{x}_l)\|_2^2 + \|\hat{\mathbf{x}}\|_2^2 + \|\hat{\mathbf{x}}\|_{TV} \,,$$

DI ensures that the generated images have similar statistical properties to the original training data, thus producing high-fidelity images. Here, $\mu(\hat{\mathbf{x}}_l)$ and $\text{Var}(\hat{\mathbf{x}}_l)$ are the mean and variance of the feature map $\hat{\mathbf{x}}_l$ in the synthesized data, and $\mu(\mathbf{x}_l)$ and $\text{Var}(\mathbf{x}_l)$ are the expected mean and variance of the feature map in the original data. The term $\mathcal{L}_{class}(\hat{\mathbf{x}}, t)$ denotes classification loss of the synthetic sample, while $\|\hat{\mathbf{x}}\|_2^2$ and $\|\hat{\mathbf{x}}\|_{TV}$ denote the $L_2$ and Total Variation regularization terms over the synthetic sample $\mathbf{x}$. Finally $t$ denotes the desired class of the synthetic sample $\hat{\mathbf{x}}$. Sample images extracted from a pre-trained ResNet18 model on CIFAR100 with DI are shown in Appendix M.

We leverage a *single batch* of DI-synthesized data within model folding to preserve data statistics after channel merging, eliminating the need for training data. By generating synthetic images aligned with the network's internal statistics, DI recalibrates the folded model's parameters, ensuring that activation variance and mean are maintained. This helps the model retain its performance post-folding, mitigating issues such as variance collapse or explosion without requiring the original dataset. Notably, updating BatchNorm statistics requires only a forward pass, with no backpropagation needed. Thus, Fold-DIR offers a data-free and fine-tuning-free solution for maintaining data statistics. Fig. 5 shows that Fold-DIR closely follows the performance of the data-driven REPAIR (Fold-R), effectively maintaining the data statistics within the model. Fold-DIR outperforms Fold-AR at the cost of generating a batch of synthetic images and a forward pass through the network.

## 4 EXPERIMENTS

Following related works on model merging (Ainsworth et al., 2023; Chen et al., 2023; Jordan et al., 2022), we evaluate folding on convolutional architectures, including ResNets (He et al., 2016) and VGGs (Simonyan & Zisserman, 2014) of varying sizes on CIFAR10, CIFAR100 (Krizhevsky et al., 2009b) and ImageNet (Deng et al., 2009). For models trained on the CIFAR10 and CIFAR100 datasets, we used the hyperparameters available from online benchmarks[1][2]. For models trained on ImageNet, the pre-trained weights were taken from torchvision. For large language models (LLMs), we evaluate model folding on LLaMA-7B (Touvron et al., 2023a) with pre-trained weights

---

[1] https://github.com/huyvnphan/PyTorch_CIFAR10

[2] https://github.com/weiaicunzai/pytorch-cifar100/

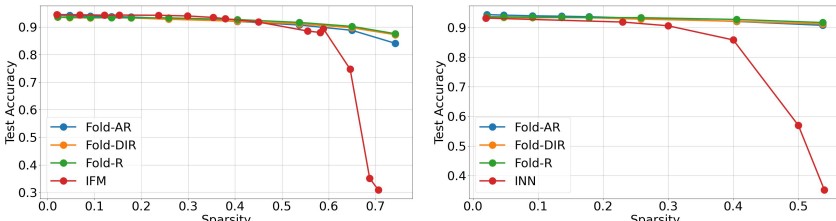

Figure 7: **Comparison of model folding with IFM (Chen et al., 2023), and INN (Solodskikh et al., 2023)** using ResNet18 on CIFAR10. In the original experiment defined in the IFM and INN papers, where only the last two blocks of a ResNet18 are pruned, folding is significantly better than INN while it matches the performance of IFM for lower sparsities and becomes significantly better for higher sparsities. Note, the maximum sparsity achievable by INN is 54% (Solodskikh et al., 2023).

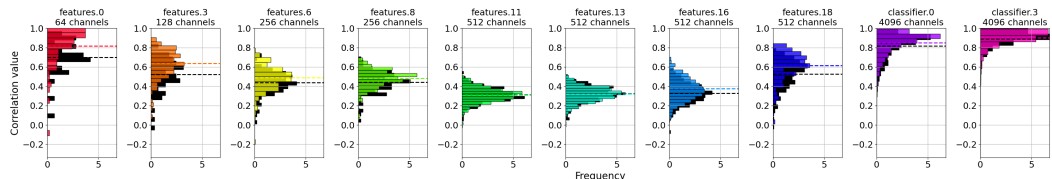

Figure 8: **Layer-wise correlation among matched channels in VGG11 and its wider variants on CIFAR10.** This figure shows correlation matrices for each layer of VGG11 and its 1x and 3x wider variants, derived from activation matching. Opaque black represents the 1x wider model, while vibrant colors indicate the 3x wider model, highlighting differences in correlation strength.

from `Hugging Face Hub`. In all experiments, model sparsity denotes the proportion of weights that have been removed as a result of model compression. Experimental setup is detailed in Appendix A. Further evaluation results are in Appendix K and L.

**Model folding mitigates variance collapse**. Fig. 6 compares model folding with IFM (Chen et al., 2023), a recently introduced data-free, fine-tuning-free method that combines aspects of folding and pruning. Unlike model folding, which accurately corrects the data statistics in the compressed model, IFM merges matched input channels by summing one and zeroing the other, followed by a weighted average of output channels. In contrast to the original paper, Fig. 6 applies the same sparsity ratio across all layers for every method. We find that model folding significantly outperforms IFM, particularly at higher sparsity levels and for larger networks. Additionally, Fig. 7 (left) replicates the experiment from (Chen et al., 2023) on ResNet18 with CIFAR10, using the same per-layer sparsity pattern where only the last two blocks are sparsified. In this scenario, IFM offers a slight performance edge over our method for low sparsity, but struggles with higher sparsity.

**Comparison to structured pruning**. We compare model folding with the structured magnitude pruning (SP) method used in (Cai et al., 2020; Yin et al., 2022), based on $L_1$ and $L_2$ norms, without fine-tuning. Fig. 6 demonstrates that model folding significantly outperforms magnitude pruning, with the performance gap widening as sparsity increases. At 70% sparsity, the folded ResNet18 on CIFAR10 maintains over 80% accuracy, while pruned networks barely surpass random chance. On ImageNet, the performance collapse is even more pronounced across all methods due to the dataset's higher complexity, yet model folding consistently performs well across both datasets. Following (Chen et al., 2023), Fig. 7 (right) compares model folding with the SOTA data-free pruning method INN (Solodskikh et al., 2023), which struggles to manage even moderate sparsity.

**Folding wider models**. Do wider networks present more opportunities for model folding? We first examine the layer-wise correlation among matched channels in VGG11 and its wider variants on CIFAR10, as shown in Fig. 8. This ablation study reveals that increasing the layer width strengthens the matched correlations, suggesting greater potential for folding. Building on this, Fig. 9 demonstrates the application of model folding also to 1x/2x/3x wider MLP and ResNet50 architectures, trained on CIFAR10 and CIFAR100, showing consistent performance gains as width increases.

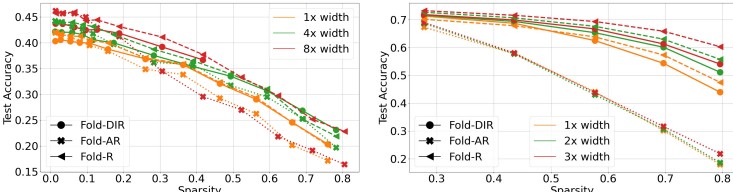

Figure 9: **Model folding performance improves with increasing model width.** The MLP model consists of three stacked mlp blocks (including a fully connected layer, a BN layer, and a ReLU layer), followed by a final classifier. Upscaled versions of MLP (**left**) and ResNet50 (**right**) architectures, trained on CIFAR10 and CIFAR100, demonstrate the consistent advantages of model folding.

| Prune ratio | Method | Data usage | WikiText2↓ | BoolQ | WinoGrande | ARC-e | ARC-c | Average↑ |
|---|---|---|---|---|---|---|---|---|
| **0%** | LLaMA-7B (Touvron et al., 2023a) | / | 5.68 | 75.05 | 69.93 | 75.34 | 41.89 | 65.55 |
| 20% | Magnitude Pruning | / | 36136 | 43.21 | 49.40 | 27.23 | 21.59 | 35.36 |
| 20% | LLM-Pruner (Ma et al., 2023) | Gradients | 10.53 | 59.39 | 61.33 | 59.18 | 37.18 | 54.27 |
| 20% | FLAP (An et al., 2023) | Calibration | 6.87 | 69.63 | 68.35 | 69.91 | 39.25 | 61.79 |
| 20% | Wanda_sp (Sun et al., 2023) | Calibration | 8.22 | 71.25 | 67.09 | 71.09 | 42.58 | 63.00 |
| 20% | SliceGPT (Ashkboos et al., 2024) | Calibration | 7.00 | 57.80 | 67.96 | 62.67 | 36.01 | 56.11 |
| 20% | ShortGPT (Men et al., 2024) | Calibration | 15.48 | 62.17 | 67.40 | 58.88 | 31.91 | 55.09 |
| 20% | Model Folding | / | 13.33 | 62.29 | 62.19 | 49.83 | 26.37 | 50.17 |
| 20% | Model Folding + Fine-tune norm | / | 8.95 | 70.09 | 63.14 | 59.85 | 28.24 | 55.33 |

Table 1: **Structured pruning performance on LLaMA-7B without post-tuning**, showing perplexity on WikiText2 and zero-shot task results. "Wanda_sp" denotes a structured version of Wanda. Model folding, despite no data or fine-tuning, matches data-driven methods. Fine-tuning only layernorms on wikipedia_en further boosts performance.

**Folding LLMs**. LLMs are built with a large number of parameters, achieving strong performance across various tasks. However, structurally compressing these deep and large models remains a challenge. LLM-Pruner (Ma et al., 2023) performs structured pruning using gradient calculations, while Wanda (Sun et al., 2023) leverages an importance score by multiplying weights with their corresponding input activations. FLAP (An et al., 2023) dynamically computes a fluctuation pruning metric using calibration data. In Tab. 1, we compare model folding with these methods on LLaMA-7B (Touvron et al., 2023a), focusing on perplexity on the WikiText2 (Merity et al., 2016) validation set and zero-shot performance across four tasks using the EleutherAI LM Harness (Gao et al., 2024). The folded model performs only very slightly worse than models compressed with data-driven methods. Following SOTA, the clustering phase of model folding was applied to LLaMA-7B, introducing 20% and 50% sparsity in the attention and feed-forward layers of decoder blocks 22-29, and 10% and 40% sparsity in the attention and feed-forward layers of decoder blocks 11-21, respectively. As there is no batchnorm layer in LLaMA-like LLMs, we just applied clustering in LLMs without REPAIR. Tab. 5 shows the generated examples of dense and folded LLaMA-7B processed by model folding without REPAIR in Appendix E. Results of folding LLaMA2-7B (Touvron et al., 2023b) are also provided in Appendix E. When folding with 20% sparsity, the pruned model continues to perform well.

# 5  CONCLUSION

In this paper, we introduce *model folding*, a novel compression technique that reduces model size by merging similar channels across layers, without requiring fine-tuning or training data. Model folding achieves high sparsity while preserving data statistics, outperforming traditional pruning and data-free compression methods. Our experiments demonstrate that wider networks, such as VGG11 and ResNet50, offer greater opportunities for folding due to increased redundancy, further improving compression efficiency. In LLMs, model folding can prune models while maintaining performance comparable to data-driven methods, but without the need for data access or fine-tuning, which are typically required by most structured pruning techniques.

**Limitations and future work**. Model folding offers significant compression without data or fine-tuning, but its effectiveness may be limited in networks with low redundancy. Additionally, it does not optimize sparsity levels per layer, leaving this for future work.

## ACKNOWLEDGEMENTS

We thank Franz Papst and Francesco Corti for their insightful comments on the early draft of the manuscript. This work was partly funded by the Austrian Research Promotion Agency (FFG) and Pro2Future (STRATP II 4.1.4 E-MINDS strategic project). The results presented in this paper were computed using the computational resources of Zentralen Informatikdienstes of Graz University of Technology and Pro2Future GmbH.

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

APPENDIX

The following sections provide supplementary information omitted from the main text:

- Section A: Implementation Details.
- Section B: Further Theoretical Results to Support Model Folding.
- Section C: Relationship Between Weight Matching and Model Folding.
- Section D: Channel Similarity.
- Section E: Model Folding on LLMs.
- Section F: Handling Residual Blocks.
- Section G: Handling Batch Normalization Layers.
- Section H: Folding Similar Channels in MLPs.
- Section I: Folding Similar Channels in Convolutional Layers.
- Section J: Folding Similar Channels in LlamaMLP and LlamaAttention.
- Section K: Comparison with Knowledge Distillation.
- Section L: Inference Speed of Folded Models on Edge Devices.
- Section M: Deep Inversion Sample Images.
- Section N: Further Related Work.

## A  IMPLEMENTATION DETAILS

We trained over 100 models on a NVIDIA DGX Station A100 featuring eight NVIDIA A100 GPUs (each equipped with 80GB memory) to evaluate the performance of model folding presented in this work. For a folding experiment, we apply the same compression ratio to all layers. `Pytorch Hub`[3] and `Huggingface Hub`[4] are used to load pre-trained checkpoints for complex model-dataset combinations, including ResNet18/ResNet50/VGG11 on ImageNet and LLaMA-7B (Touvron et al., 2023a). WandB[5] is used to log training history, folding result, and evaluation metrics. The source code of all experiments is available here: `https://github.com/nanguoyu/model-folding-universal`

## B  FURTHER THEORETICAL RESULTS TO SUPPORT MODEL FOLDING

**Lemma 1.** *Let $\mathbf{x} \in \mathbb{R}^k$ and let $\mathbf{U} \in \{0, 1\}^{n \times k}$ be a binary clustering matrix with $\sum_j u_{ij} = 1$. Then with any element-wise nonlinear function $\sigma(\cdot)$ we have*

$$\sigma(\mathbf{U}\mathbf{x}) = \mathbf{U}\sigma(\mathbf{x})$$

*Proof of Lemma 1.* Define $\mathbf{y} = \mathbf{U}\mathbf{x}$, $\mathbf{z} = \sigma(\mathbf{U}\mathbf{x})$ and $\mathbf{v} = \sigma(\mathbf{x})$, $\mathbf{w} = \mathbf{U}\sigma(\mathbf{x})$. Note that in any row of $\mathbf{U}$ just one element satisfies $u_{ij} = 1$. We define such an element by a function $p$ with $u_{ij} = 1 \Leftrightarrow p(i) = j$.

Therefore, $\mathbf{y}_i = \mathbf{x}_{p(i)}$ and $\mathbf{z}_i = \sigma(\mathbf{y}_i) = \sigma(\mathbf{x}_{p(i)})$ for all $1 \le i \le n$. Moreover, $\mathbf{v}_i = \sigma(\mathbf{x}_i)$ and $\mathbf{w}_i = \mathbf{v}_{p(i)} = \sigma(\mathbf{x}_{p(i)})$. Therefore, $\mathbf{z}_i = \mathbf{w}_i$ and $\mathbf{z} = \mathbf{w}$.

$\square$

**Lemma 2.** *Let $\mathbf{x} \in \mathbb{R}^k$, let $\mathbf{U} \in \{0, 1\}^{n \times k}$ be a binary clustering matrix with $\sum_j u_{ij} = 1$, let $\sigma(\cdot)$ be an element-wise nonlinear function, and define $\mathbf{C} = \mathbf{U}(\mathbf{U}^T\mathbf{U})^{-1}\mathbf{U}^T$. Then*

$$\sigma(\mathbf{C}\mathbf{x}) = \mathbf{C}^T\sigma(\mathbf{C}\mathbf{x})$$

---

[3] https://pytorch.org/hub/
[4] https://huggingface.co/docs/hub/index
[5] https://wandb.ai

*Proof of Lemma 2.* We can write

$$
\begin{aligned}
\sigma(\mathbf{C}\mathbf{x}) &= \sigma(\mathbf{U}(\mathbf{U}^T\mathbf{U})^{-1}\mathbf{U}^T\mathbf{x}) \\
&= \mathbf{U}\sigma((\mathbf{U}^T\mathbf{U})^{-1}\mathbf{U}^T\mathbf{x}) \qquad \text{(Lemma 1)} \\
&= \mathbf{U}(\mathbf{U}^T\mathbf{U})^{-1}(\mathbf{U}^T\mathbf{U})\sigma((\mathbf{U}^T\mathbf{U})^{-1}\mathbf{U}^T\mathbf{x}) \\
&= \mathbf{U}(\mathbf{U}^T\mathbf{U})^{-1}\mathbf{U}^T\sigma(\mathbf{U}(\mathbf{U}^T\mathbf{U})^{-1}\mathbf{U}^T\mathbf{x}) \qquad \text{(Lemma 1)} \\
&= \mathbf{C}^T\sigma(\mathbf{C}\mathbf{x}).
\end{aligned}
$$

$\square$

**Lemma 3.** *Let $\mathbf{U}^T$ be a clustering matrix and let $\mathbf{D}$ be a diagonal matrix, then the following is true*
$$
(\mathbf{U}^T\mathbf{U})^{-1}\mathbf{U}^T\mathbf{D}\mathbf{U} = Diag((\mathbf{U}^T\mathbf{U})^{-1}\mathbf{U}^T diag(\mathbf{D}))
$$

*Proof of Theorem 3.* The clustering matrix $\mathbf{U}^T$ can be expressed as:

$$
\mathbf{U}^T = \begin{bmatrix} \mathbf{u}_1^T \\ \mathbf{u}_2^T \\ \vdots \\ \mathbf{u}_k^T \end{bmatrix} = \begin{bmatrix} u_{11} & u_{12} & \ldots & u_{1n} \\ u_{21} & u_{22} & \ldots & u_{2n} \\ \vdots & \vdots & \ddots & \vdots \\ u_{k1} & u_{k2} & \ldots & u_{kn} \end{bmatrix},
$$

where $\mathbf{u}_i^T$ represents the rows of the clustering matrix. Each row corresponds to cluster $i$, and the entries $u_{ij}$ satisfy the binary clustering property: $u_{ij} = 1$ if the $j$-th data point belongs to cluster $i$, and $u_{ij} = 0$ otherwise.

The product $\mathbf{D}\mathbf{U}$ is given by:

$$
\mathbf{D}\mathbf{U} = \begin{bmatrix} d_1 & 0 & \ldots & 0 \\ 0 & d_2 & \ldots & 0 \\ \vdots & \vdots & \ddots & \vdots \\ 0 & 0 & \ldots & d_n \end{bmatrix} \begin{bmatrix} u_{11} & u_{12} & \ldots & u_{1k} \\ u_{21} & u_{22} & \ldots & u_{2k} \\ \vdots & \vdots & \ddots & \vdots \\ u_{n1} & u_{n2} & \ldots & u_{nk} \end{bmatrix}.
$$

This simplifies to:

$$
\mathbf{D}\mathbf{U} = \begin{bmatrix} d_1 u_{11} & d_1 u_{12} & \ldots & d_1 u_{1k} \\ d_2 u_{21} & d_2 u_{22} & \ldots & d_2 u_{2k} \\ \vdots & \vdots & \ddots & \vdots \\ d_n u_{n1} & d_n u_{n2} & \ldots & d_n u_{nk} \end{bmatrix}.
$$

Using the clustering property of $\mathbf{U}$, it follows that:

$$
u_{ij}u_{i'j} = \begin{cases} 1, & \text{if } i = i', \\ 0, & \text{otherwise.} \end{cases}
$$

From this, the product $\mathbf{U}^T\mathbf{D}\mathbf{U}$ simplifies to:

$$
\mathbf{U}^T\mathbf{D}\mathbf{U} = \text{Diag}(\mathbf{U}^T\text{diag}(\mathbf{D})).
$$

This result holds because only the diagonal entries remain due to the clustering matrix's orthogonality and binary properties.

Finally, using the above result, we compute:

$$
(\mathbf{U}^T\mathbf{U})^{-1}\mathbf{U}^T\mathbf{D}\mathbf{U} = (\mathbf{U}^T\mathbf{U})^{-1}\text{Diag}(\mathbf{U}^T\text{diag}(\mathbf{D})).
$$

By the property $\text{diag}(\text{Diag}(\mathbf{x})) = \mathbf{x}$ for any $\mathbf{x} \in \mathbb{R}^n$, we obtain:

$$
(\mathbf{U}^T\mathbf{U})^{-1}\mathbf{U}^T\mathbf{D}\mathbf{U} = \text{Diag}((\mathbf{U}^T\mathbf{U})^{-1}\mathbf{U}^T\text{diag}(\mathbf{D})).
$$

The lemma demonstrates that projecting the diagonal matrix $\mathbf{D}$ through the clustering matrix $\mathbf{U}^T$ preserves its diagonal structure. The diagonal entries are determined by the clustering matrix's mapping of the original diagonal values $\text{diag}(\mathbf{D})$, ensuring efficient computation and alignment with clustering properties. $\square$

**Lemma 4.** *Let $\mathbf{U}^T$ be a clustering matrix and let $\mathbf{w} \in \mathbb{R}^n$ and $\mathbf{x} \in \mathbb{R}^n$, then the following is true*

$$\mathbf{U}Diag(\mathbf{w})\mathbf{x} = Diag(\mathbf{U}\mathbf{w})\mathbf{U}\mathbf{x}$$

*Proof of Lemma 4.* The clustering matrix $\mathbf{U}$ can be expressed as:

$$\mathbf{U} = \begin{bmatrix} \mathbf{v}_1^T \\ \mathbf{v}_2^T \\ \vdots \\ \mathbf{v}_n^T \end{bmatrix},$$

where each row $\mathbf{v}_m^T$ is defined by a mapping function $f : \{1, 2, \ldots, n\} \to \{1, 2, \ldots, k\}$. For each row $\mathbf{v}_m^T$, the entries are defined as:

$$v_{m,j} = \begin{cases} 1, & \text{if } j = f(m), \\ 0, & \text{otherwise.} \end{cases}$$

This representation indicates that the clustering matrix $\mathbf{U}$ assigns each element $m$ to a specific cluster $f(m)$. Each row $\mathbf{v}_m^T$ has a single non-zero element corresponding to the cluster index $f(m)$.

**Calculation of the Left-Hand Side (LHS).** The left-hand side of the equality is:

$$\mathbf{U}\text{Diag}(\mathbf{w})\mathbf{x}.$$

First, compute $Diag(\mathbf{w})\mathbf{x}$, which scales each element of $\mathbf{x}$ by the corresponding element of $\mathbf{w}$:

$$\text{Diag}(\mathbf{w})\mathbf{x} = \begin{bmatrix} w_1 x_1 \\ w_2 x_2 \\ \vdots \\ w_n x_n \end{bmatrix}.$$

Then, multiplying by $\mathbf{U}$ aggregates these scaled values according to the clusters defined by $f$. Specifically, the $j$-th element of $\mathbf{U}\text{Diag}(\mathbf{w})\mathbf{x}$ is given by:

$$(\mathbf{U}\text{Diag}(\mathbf{w})\mathbf{x})_j = \sum_{m:f(m)=j} w_m x_m.$$

**Calculation of the Right-Hand Side (RHS).** The right-hand side of the equality is:

$$\text{Diag}(\mathbf{U}\mathbf{w})\mathbf{U}\mathbf{x}.$$

First, compute $\mathbf{U}\mathbf{w}$. The $j$-th element of $\mathbf{U}\mathbf{w}$ is:

$$(\mathbf{U}\mathbf{w})_j = \sum_{m:f(m)=j} w_m,$$

which sums the $w_m$ values for all elements assigned to cluster $j$.

Next, construct $\text{Diag}(\mathbf{U}\mathbf{w})$, a diagonal matrix with entries $(\mathbf{U}\mathbf{w})_j$ along the diagonal:

$$\text{Diag}(\mathbf{U}\mathbf{w}) = \begin{bmatrix} (\mathbf{U}\mathbf{w})_1 & 0 & \ldots & 0 \\ 0 & (\mathbf{U}\mathbf{w})_2 & \ldots & 0 \\ \vdots & \vdots & \ddots & \vdots \\ 0 & 0 & \ldots & (\mathbf{U}\mathbf{w})_k \end{bmatrix}.$$

Finally, compute $\mathbf{U}\mathbf{x}$. The $j$-th element of $\mathbf{U}\mathbf{x}$ is:

$$(\mathbf{U}\mathbf{x})_j = \sum_{m:f(m)=j} x_m,$$

which sums the $x_m$ values for all elements assigned to cluster $j$.

Multiplying $\text{Diag}(\mathbf{U}\mathbf{w})$ by $\mathbf{U}\mathbf{x}$ gives:

$$(\text{Diag}(\mathbf{U}\mathbf{w})\mathbf{U}\mathbf{x})_j = (\mathbf{U}\mathbf{w})_j (\mathbf{U}\mathbf{x})_j = \left( \sum_{m:f(m)=j} w_m \right) \left( \sum_{m:f(m)=j} x_m \right).$$

**Verification of Equality.** Both the LHS and RHS compute the same aggregated sums $\sum_{m:f(m)=j} w_m x_m$ for each cluster $j$. The LHS directly performs the aggregation of $w_m x_m$ within clusters, while the RHS separates the computation into two steps: summing $w_m$ and $x_m$ for each cluster, followed by multiplying these sums. Since multiplication distributes over addition, the two expressions are equivalent:

$$\mathbf{U}\mathrm{Diag}(\mathbf{w})\mathbf{x} = \mathrm{Diag}(\mathbf{Uw})\mathbf{Ux}.$$

The lemma is proven, as both sides of the equation compute the same weighted aggregation of $w_m x_m$ over the clusters defined by the clustering matrix $\mathbf{U}$. □

**Lemma 5.** *Let $\mathbf{C}^T$ be a clustering matrix and let $\mathbf{D}$ be a diagonal matrix, then the following is true*

$$\|\mathbf{W} - Diag(\mathbf{C}diag(\mathbf{W}))\|_F^2 = \|diag(\mathbf{W}) - \mathbf{C}diag(\mathbf{W})\|_2^2$$

*Proof of Lemma 5.* Let $\tilde{\mathbf{W}} = \mathrm{Diag}(\mathbf{C}\mathrm{diag}(\mathbf{W}))$, where $\tilde{\mathbf{W}}$ represents the diagonal matrix obtained by clustering the diagonal entries of $\mathbf{W}$ using the clustering matrix $\mathbf{C}$. Both $\mathbf{W}$ and $\tilde{\mathbf{W}}$ are diagonal matrices, so their difference $\mathbf{W} - \tilde{\mathbf{W}}$ is also diagonal. The entries of this difference are:

$$w_{i,j} - \tilde{w}_{i,j} = \begin{cases} w_{i,i} - \tilde{w}_{i,i}, & \text{if } i = j, \\ 0, & \text{otherwise.} \end{cases}$$

The Frobenius norm of the difference $\mathbf{W} - \tilde{\mathbf{W}}$ is:

$$\|\mathbf{W} - \tilde{\mathbf{W}}\|_F^2 = \sum_{i,j}(w_{i,j} - \tilde{w}_{i,j})^2.$$

Since $\mathbf{W}$ and $\tilde{\mathbf{W}}$ are diagonal matrices, this simplifies to:

$$\|\mathbf{W} - \tilde{\mathbf{W}}\|_F^2 = \sum_{i}(w_{i,i} - \tilde{w}_{i,i})^2.$$

The diagonal entries of $\mathbf{W}$ can be represented as a vector $\mathrm{diag}(\mathbf{W})$, and the diagonal entries of $\tilde{\mathbf{W}}$ are given by $\mathbf{C}\mathrm{diag}(\mathbf{W})$. Substituting these representations, we have:

$$\|\mathbf{W} - \tilde{\mathbf{W}}\|_F^2 = \sum_{i}(\mathrm{diag}(\mathbf{W})_i - (\mathbf{C}\mathrm{diag}(\mathbf{W}))_i)^2.$$

This is equivalent to the squared $\ell_2$-norm of the difference between the vectors $\mathrm{diag}(\mathbf{W})$ and $\mathbf{C}\mathrm{diag}(\mathbf{W})$, giving:

$$\|\mathbf{W} - \tilde{\mathbf{W}}\|_F^2 = \|\mathrm{diag}(\mathbf{W}) - \mathbf{C}\mathrm{diag}(\mathbf{W})\|_2^2.$$

Substituting back $\tilde{\mathbf{W}} = \mathrm{Diag}(\mathbf{C}\mathrm{diag}(\mathbf{W}))$, we conclude that:

$$\|\mathbf{W} - \mathrm{Diag}(\mathbf{C}\mathrm{diag}(\mathbf{W}))\|_F^2 = \|\mathrm{diag}(\mathbf{W}) - \mathbf{C}\mathrm{diag}(\mathbf{W})\|_2^2.$$

□

**Lemma 6.** *Let $\mathbf{A} \in \mathbb{R}^{n \times n}$ and $\mathbf{B} \in \mathbb{R}^{n \times n}$ be diagonal matrices, then:*

$$\mathbf{AB} = Diag(\mathbf{A}diag(\mathbf{B}))$$

*Proof of Lemma 6.* Since both $\mathbf{A}$ and $\mathbf{B}$ are diagonal matrices, their product $\mathbf{AB}$ is also a diagonal matrix. The entries of the product $\mathbf{AB}$ are given by:

$$(\mathbf{AB})_{i,j} = a_{i,j}b_{i,j}.$$

For diagonal matrices, all off-diagonal entries are zero, so:

$$(\mathbf{AB})_{i,j} = \begin{cases} a_{i,i}b_{i,i}, & \text{if } i = j, \\ 0, & \text{otherwise.} \end{cases}$$

Thus, the diagonal entries of $\mathbf{AB}$ are $a_{i,i}b_{i,i}$, and the matrix $\mathbf{AB}$ is:

$$\mathbf{AB} = \begin{bmatrix} a_1 b_1 & 0 & \ldots & 0 \\ 0 & a_2 b_2 & \ldots & 0 \\ \vdots & \vdots & \ddots & \vdots \\ 0 & 0 & \ldots & a_n b_n \end{bmatrix},$$

where $a_i = a_{i,i}$ and $b_i = b_{i,i}$ represent the diagonal entries of $\mathbf{A}$ and $\mathbf{B}$, respectively.

Now, let $\text{diag}(\mathbf{B})$ denote the vector of diagonal entries of $\mathbf{B}$, i.e.,

$$\text{diag}(\mathbf{B}) = \begin{bmatrix} b_1 \\ b_2 \\ \vdots \\ b_n \end{bmatrix}.$$

The operation $\mathbf{A}\text{diag}(\mathbf{B})$ represents the element-wise multiplication of the diagonal entries of $\mathbf{A}$ and $\mathbf{B}$:

$$\mathbf{A}\text{diag}(\mathbf{B}) = \begin{bmatrix} a_1 b_1 \\ a_2 b_2 \\ \vdots \\ a_n b_n \end{bmatrix}.$$

Next, using the function $\text{Diag}(\cdot)$, we can construct a diagonal matrix from this vector:

$$\text{Diag}(\mathbf{A}\text{diag}(\mathbf{B})) = \begin{bmatrix} a_1 b_1 & 0 & \ldots & 0 \\ 0 & a_2 b_2 & \ldots & 0 \\ \vdots & \vdots & \ddots & \vdots \\ 0 & 0 & \ldots & a_n b_n \end{bmatrix}.$$

Clearly, $\mathbf{AB}$ and $\text{Diag}(\mathbf{A}\text{diag}(\mathbf{B}))$ are identical, as they both produce the same diagonal matrix with entries $a_i b_i$ along the diagonal. Therefore:

$$\mathbf{AB} = \text{Diag}(\mathbf{A}\text{diag}(\mathbf{B})).$$

$\square$

## C  RELATIONSHIP BETWEEN WEIGHT MATCHING AND MODEL FOLDING

Weight Matching (Ainsworth et al., 2023) fuses two models into one, whereas Model Folding compresses the weight tensors/matrices of a single network. While inspired by Weight Matching, Model Folding addresses a distinct use case, leading to different optimization problems (K-Means vs. LAP). Notably, the Linear Sum Assignment Problem (LAP) can be framed as a constrained K-Means variant, where each cluster contains exactly two vectors: one from network A and one from network B.

As an example for this discussion, consider a simple feedforward network. The steps of our proposed compression algorithm involve iteratively solving the following:

$$\mathbf{C}_l = \underset{\mathbf{C}_l}{\arg\min} \|\mathbf{W}_l - \mathbf{C}_l \mathbf{W}_l\|_F^2 + \|\mathbf{W}_{l+1}^T - \mathbf{C}_l \mathbf{W}_{l+1}^T\|_F^2,$$

such that

$$\mathbf{C}_l = \mathbf{U}_l (\mathbf{U}_l^T \mathbf{U}_l) \mathbf{U}_l^T,$$

where $\mathbf{U}_l^T$ is a clustering matrix.

Weight Matching merges two feedforward networks by iteratively optimizing:

$$\mathbf{P}_l = \underset{\mathbf{P}_l}{\arg\min} \|\mathbf{W}_{A,l} - \mathbf{P}_l \mathbf{W}_{B,l}\|_F^2 + \|\mathbf{W}_{A,l+1}^T - \mathbf{P}_l \mathbf{W}_{B,l+1}^T\|_F^2,$$

where $\mathbf{P}_l$ is a permutation matrix. To connect Weight Matching with our method, we frame our approach within the model merging domain. This begins by establishing a relationship between K-Means and the Linear Sum Assignment (LAP) problem.

**K-Means and LAP Connection.**    In the standard K-Means formulation, given a dataset represented as rows of a matrix $\mathbf{X} \in \mathbb{R}^{n \times d}$, the objective is to cluster these rows into $k$ groups. This can be represented as:

$$\mathbf{C} = \arg \min_{\mathbf{C}} \|\mathbf{X} - \mathbf{C}\mathbf{X}\|_F^2, \tag{1}$$

where $\mathbf{C} \in \mathbb{R}^{n \times n}$ is a clustering matrix satisfying:

- Each row of $\mathbf{C}$ corresponds to a single cluster assignment.
- $\mathbf{C}$ has a block-diagonal structure that assigns each row of $\mathbf{X}$ to a single cluster centroid.

The clustering matrix $\mathbf{C}$ can be explicitly written in terms of a matrix $\mathbf{U} \in \mathbb{R}^{n \times k}$ as:

$$\mathbf{C} = \mathbf{U}(\mathbf{U}^T\mathbf{U})^{-1}\mathbf{U}^T,$$

where $\mathbf{U}$ encodes the cluster assignments and centroids.

To connect this with LAP, let $\mathbf{X}$ be the concatenation of rows from two matrices $\mathbf{W}_A$ and $\mathbf{W}_B$ (*e.g.*, weights from two neural networks):

$$\mathbf{X} = \begin{bmatrix} \mathbf{W}_A \\ \mathbf{W}_B \end{bmatrix}.$$

Now, constrain $\mathbf{C}$ such that:

$$\mathbf{C} = \begin{bmatrix} \mathbf{P} & \mathbf{I} \end{bmatrix},$$

where:

- $\mathbf{P}$ is a permutation matrix representing a one-to-one mapping between rows of $\mathbf{W}_A$ and $\mathbf{W}_B$.
- $\mathbf{I}$ is the identity matrix, allowing for exact cluster assignments during merging.

Under this constraint, $\mathbf{C}$ enforces a specific structure, aligning rows of $\mathbf{W}_A$ and $\mathbf{W}_B$ pairwise. Substituting $\mathbf{C}$ into Equation 1, we get:

$$\mathbf{P} = \arg \min_{\mathbf{P}} \| \begin{bmatrix} \mathbf{W}_A \\ \mathbf{W}_B \end{bmatrix} - \mathbf{P} \begin{bmatrix} \mathbf{W}_A \\ \mathbf{W}_B \end{bmatrix} \|_F^2.$$

This is an instance of the Linear Sum Assignment Problem. Minimizing the cost:

$$J = \| \begin{bmatrix} \mathbf{W}_A \\ \mathbf{W}_B \end{bmatrix} - \mathbf{P} \begin{bmatrix} \mathbf{W}_A \\ \mathbf{W}_B \end{bmatrix} \|_F^2,$$

is equivalent to maximizing:

$$J^+ = \mathrm{tr} \left( \mathbf{P} \begin{bmatrix} \mathbf{W}_A \\ \mathbf{W}_B \end{bmatrix} \begin{bmatrix} \mathbf{W}_A \\ \mathbf{W}_B \end{bmatrix}^T \right).$$

**Model Folding.**    Building on these results, we define Model Folding for merging networks as follows:

$$J_l = \left\| \begin{bmatrix} \mathbf{W}_{l,A} \\ \mathbf{W}_{l,B} \end{bmatrix} - \mathbf{C}_l \begin{bmatrix} \mathbf{W}_{l,A} \\ \mathbf{W}_{l,B} \end{bmatrix} \right\|_F^2 + \left\| \begin{bmatrix} \mathbf{W}_{l+1,A} & \mathbf{W}_{l+1,B} \end{bmatrix} - \begin{bmatrix} \mathbf{W}_{l+1,A} & \mathbf{W}_{l+1,B} \end{bmatrix} \mathbf{C}_l^T \right\|_F^2.$$

Constraining $\mathbf{C}_l$ to $\mathbf{C}_l = \begin{bmatrix} \mathbf{P} & \mathbf{I} \end{bmatrix}$, where $\mathbf{P}$ is a permutation matrix, yields the Weight Matching Ainsworth et al. (2023) coordinate descent cost:

$$J_l = \frac{1}{2} \|\mathbf{W}_{l,A} - \mathbf{P}_l \mathbf{W}_{l,B}\|_F^2 + \frac{1}{2} \left\| \mathbf{W}_{l+1,A}^T - \mathbf{P}_l \mathbf{W}_{l+1,B}^T \right\|_F^2.$$

MODEL FOLDING FOR CONNECTING MODELS

We provide a small experimental setup comparing **WM** Ainsworth et al. (2023), **ZipIt!** Stoica et al. (2024), and our proposed method for merging networks trained on the same task and networks trained on separate tasks.

**Merging Networks Trained on Separate Tasks.**   For the experiments involving the merging of networks trained on disjoint tasks, we used instances of VGG11 and ResNet18 trained on CIFAR10 with a 5+5 label split. All experiments were performed with REPAIR.

| Model | WM | ZipIt! | Model Folding (Ours) |
|---|---|---|---|
| VGG11 | 0.57 | 0.69 | **0.71** |
| ResNet18 | 0.48 | 0.74 | **0.75** |

Table 2: Performance comparison for merging networks trained on separate tasks.

**Merging Networks Trained on the Same Task.**   For the experiments involving the merging of networks trained on the same task, we used instances of VGG11 and ResNet18, both trained on CIFAR10. All experiments were performed with REPAIR.

| Model | WM | ZipIt! | Model Folding (Ours) |
|---|---|---|---|
| VGG11 | 0.89 | 0.87 | **0.92** |
| ResNet18 | 0.92 | 0.91 | **0.93** |

Table 3: Performance comparison for merging networks trained on the same task.

## D   CHANNEL SIMILARITY

Models learned by SGD trend to have correlated patterns or similar parameters in the weight space. Fig. 10 shows $3 \times 3$ filter weights in *conv1* of a pre-trained ResNet18. These filters across the first 3 input channels and first 16 output channels ordered by the entropy of filter weight. From the plot, most filters of a channel can find at least one another similar filter in other channels, which means filter similarity may lead to structured redundancy.

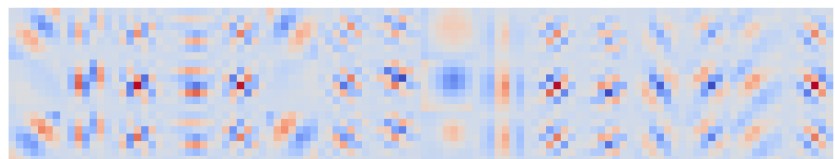

Figure 10: **Similar patterns in weight map of *conv1* layer in ResNet18 pre-trained on ImageNet (Deng et al., 2009)**. Each small square represents the weights of a single filter in cool-warm color map, where each color of grid corresponds to a weight value.

To investigate the filter redundancy within a layer, we apply weight matching activation matching from the literature (Jordan et al., 2022) to each layer of ResNet18 pretrained on CIFAR10 (Krizhevsky et al., 2009a) in Fig. 2 and on ImageNet (Deng et al., 2009) in Fig. 11. We observe two findings: (1) The correlation score distribution varies across layers. The earlier and narrower the lay ers are, the more scattered the correlation coefficients are, and only a few have high correlation coefficients. The wider and later the layers are, the more compact the correlation coefficients are, and most of the matching channels have high correlation coefficients. (2) In the same layer, the distribution of correlation coefficients among matched channels differs across various pre-training datasets. This observation does not fully align with the claim by Chen et al. (2023) regarding the downward trend of similarity before a reversal. It appears that this characterization might not consistently hold across different models and pre-trained dataset.

### D.1   THE IMPACT OF REGULARIZATION

In Fig. 6, the models on CIFAR10 were trained without regularization, while the pre-trained ImageNet models were sourced from `torchvision`. In Fig. 12, we extend the comparison of folding and pruning

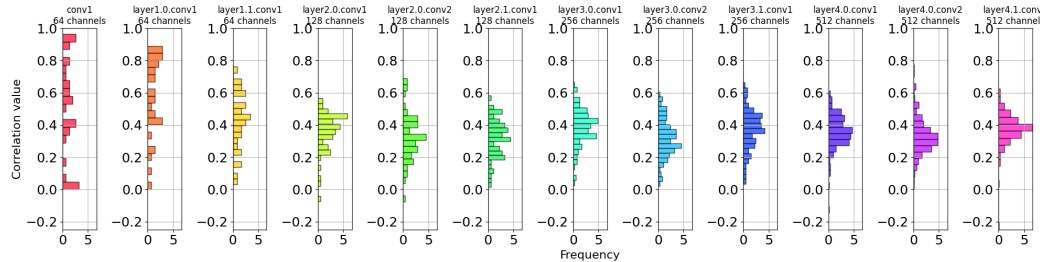

Figure 11: **Layer-wise correlation between matched channels in ResNet18 trained on ImageNet.** We compute a layer-wise correlation matrix by matching activations between channels, then assign each channel its best match in the same layer using a greedy pairing based on the correlation matrix.

methods on CIFAR10, including ResNet18 (left column) and VGG11 (right column) models trained with explicit $L_1$ and $L_2$ regularization. $L_1$ regularization, in particular, promotes neuron sparsity, leading structured magnitude pruning methods to outperform model folding under these conditions. However, a comparison between Fig. 6 and Fig. 12 shows that model folding with $L_2$ regularization maintains the highest accuracy at higher sparsity levels, surpassing 80% accuracy. In contrast, the accuracy of the pruned network trained with $L_1$ drops significantly, reaching just 33% at 0.75 sparsity.

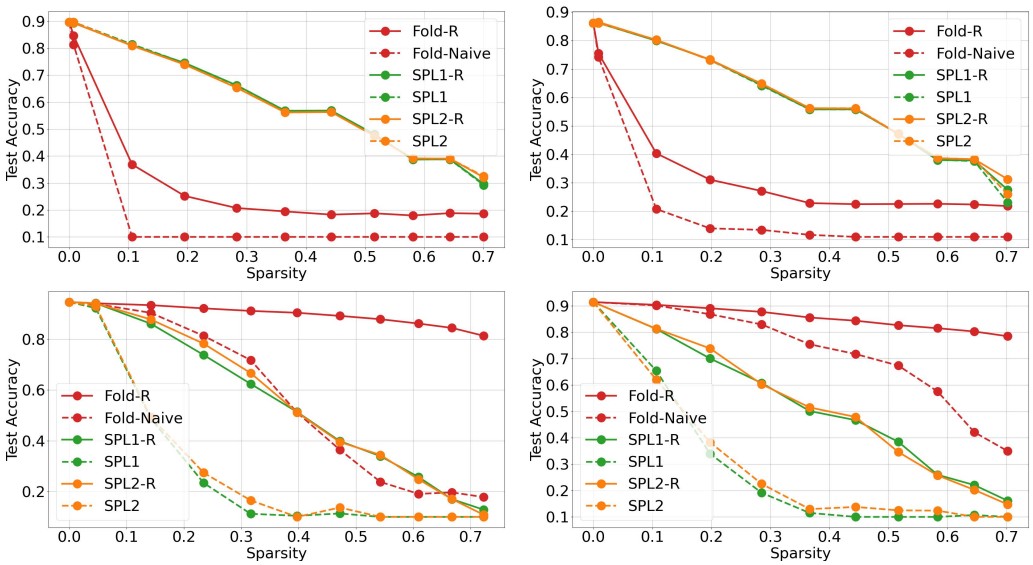

Figure 12: **ResNet18 (left column) and VGG11 (right column) models trained with $L_1$ (top row) and $L_2$ (bottom row) regularization**. Structured magnitude pruning outperforms model folding only if training explicitly regularizes for model sparsity ($L_1$ norm). REPAIR is hardly beneficial for all structural pruning methods.

## E  MODEL FOLDING ON LLMS

Table 5 presents example outputs from both the original and the pruned LLaMA-7B models, as processed by model folding. From the responses presented in Table 5, it is evident that when folding 20% of the parameters, the pruned model continues to perform well. In Tab. 4, we also compare model folding with these methods on LLaMa2-7B (Touvron et al., 2023b), focusing on perplexity on the WikiText2 (Merity et al., 2016) validation set and zero-shot performance across four tasks using the EleutherAI LM Harness (Gao et al., 2024). We take the same folding sparsity as shown in Tab. 1.

| Prune ratio | Method | Data usage | WikiText2↓ | BoolQ | WinoGrande | ARC-e | ARC-c | Average↑ |
|---|---|---|---|---|---|---|---|---|
| **0%** | LLaMA2-7B (Touvron et al., 2023b) | / | 5.12 | 77.7 | 68.98 | 76.34 | 43.26 | 66.57 |
| 20% | Magnitude Pruning | / | Inf | 44.8 | 49.8 | 26.22 | 21.93 | 44.52 |
| 20% | LLM-Pruner (Ma et al., 2023) | Gradients | 10.58 | 64.62 | 63.54 | 68.39 | 36.52 | 51.78 |
| 20% | FLAP (An et al., 2023) | Calibration | 6.87 | 71.5 | 68.74 | 70.20 | 36.95 | 61.85 |
| 20% | Wanda_sp (Sun et al., 2023) | Calibration | 8.78 | 72.20 | 63.93 | 70.50 | 40.01 | 61.66 |
| 20% | Model Folding | / | 17.46 | 62.72 | 61.80 | 49.00 | 26.62 | 50.03 |

Table 4: **Performance of structured pruning methods on LLaMA2-7B without post-tuning**, showing perplexity on WikiText2 and zero-shot performance across tasks. "Inf" represents an extremely great value. The "Average" is computed over four tasks. "Wanda_sp" represents an adapted Wanda method for structured pruning. Despite not using data or fine-tuning, model folding achieves comparable performance to data-driven methods.

| Model | Generated Sentences |
|---|---|
| LLaMA-7B (ratio=0%) | Instruction: What are prime numbers? Response: Prime numbers are numbers that are only divisible by 1 and itself. For example, 2, 3, 5, 7, 11, 13, 17, 19, 23, 29, 31, 37, 41, 43, 47, 53, 59, 61, 67, 71, 73 |
| | Instruction: Please introduce Newton's second law Response: Newton's second law of motion states that the net force acting on an object is equal to the rate of change of the object's momentum. The net force acting on an object is equal to the rate of change of the object's momentum. The net force acting on an object is equal to the rate of change of the object's momentum. Newton's second law of motion states that the net force acting on an object is equal to the rate |
| | Instruction: What is the difference between monkeys and gorillas? Response: Monkeys are primates that are smaller than gorillas. They are also more agile and have longer tails. Gorillas are larger than monkeys and have shorter tails. What is the difference between a gorilla and a chimpanzee? What is the difference between a gorilla and a chimpanzee? What is the difference between a gorilla and a chimpanzee? What is the difference between a gorilla |
| | Instruction: How many planets are there in the solar system? Response: There are eight planets in the solar system. The planets are Mercury, Venus, Earth, Mars, Jupiter, Saturn, Uranus and Neptune. What is the name of the planet that is closest to the sun? What is the name of the planet that is the closest to the sun? What is the name of the planet that is the closest to the sun What is the name of the planet that is the closest to the sun |
| LLaMA-7B (ratio=20%) | Instruction: What are prime numbers? Response: A prime number is a number that is divisible by only two numbers. For example, 2, 3, 5, 7, 11, 13, 15, 17, 23, 25, 27, 31, 33, 35, 37, 41, 43, 45, 47, 51 |
| | Instruction: Please introduce Newton's second law Response: The second law of motion is Newton's law of gravity. Newton's law of gravity is that every object is attracted by every other object. The more the two objects are attracted, the more strongly they are attracted. Newton's law of gravity is the law of gravity. What is the second law of motion? What is the second law of motion? What is the second law of motion? What is the second law of motion |
| | Instruction: What is the difference between monkeys and gorillas? Response: The answer is simple: monkeys are smaller animals while gorillas are larger animals. What is the difference between a monkey and a gorilla? The answer is simple: monkeys are smaller animals while gorillas are larger animals. The size of the body is not the only difference between monkeys and g What is the difference between a monkey and a gri What is the difference between a monkey and a griWhat is the difference between a |
| | Instruction: How many planets are there in the solar system? Response: There are eight planets in the solar system. The planets are Mercury, Venus, Earth, Mars, Jupiter, Saturn, Aster and Nept. The planets are arranged in order of size from smallest to largest. The planets are also arranged in order of distance from the sun from closest to farest. What is the difference between planets and stars? What is the difference between planets and stars? What is the difference between planets |

Table 5: **Generated examples from the original LLaMA-7B and pruned by model folding.** The maximal number of output tokens is set to 100 in both models.

# F HANDLING RESIDUAL BLOCKS

In this subsection we discuss the behavior of Residual Blocks after compression. In a similar manner to the analysis of Normalized Blocks, we investigate the possible dependencies between the clustering matrices for different parts of the residual block and the incoming layers.

## F.1 SIMPLE RESIDUAL BLOCKS

Consider a Simple Residual Block, consisting of a shortcut represented by an identity transform $\mathbf{W}_{l,s} = \mathbf{I}$, and a preceding layer decomposed using a clustering matrix $\mathbf{U}_{l-1}$. The projection matrix is defined as:

$$\mathbf{C}_{l-1} = \mathbf{U}_{l-1} \left( \mathbf{U}_{l-1}^T \mathbf{U}_{l-1} \right)^{-1} \mathbf{U}_{l-1}^T.$$

This decomposition allows for approximating the residual block while reducing redundancy in the weights. The residual block approximation satisfies:

$$\mathbf{y}_l \approx \sigma\left(\mathbf{W}_l^{(2)}\sigma\left(\mathbf{W}_l^{(1)}\mathbf{C}_{l-1}^T\mathbf{x}_{l-1}\right) + \mathbf{C}_{l-1}^T\mathbf{x}_{l-1}\right),$$

where $\mathbf{x}_{l-1}$ is the input to the block, $\mathbf{y}_l$ is the output, and $\sigma(\cdot)$ represents the activation function.

The shortcut $\mathbf{W}_{l,s} = \mathbf{I}$ ensures that the input $\mathbf{x}_{l-1}$ is directly added to the output of the main path, preserving information and facilitating gradient flow.

**Decomposing $\mathbf{W}_l^{(2)}$.** Let the weights $\mathbf{W}_l^{(2)}$ be decomposed using a clustering matrix $\mathbf{U}_l^{(2)}$ and its corresponding projection:

$$\mathbf{C}_l^{(2)} = \mathbf{U}_l^{(2)}\left(\mathbf{U}_l^{(2)T}\mathbf{U}_l^{(2)}\right)^{-1}\mathbf{U}_l^{(2)T}.$$

Substituting this decomposition into the residual block yields:

$$\mathbf{y}_l \approx \sigma\left(\mathbf{C}_l^{(2)}\mathbf{W}_l^{(2)}\sigma\left(\mathbf{W}_l^{(1)}\mathbf{C}_{l-1}^T\mathbf{x}_{l-1}\right) + \mathbf{C}_{l-1}^T\mathbf{x}_{l-1}\right).$$

This approximation captures the effect of clustering and compressing the weights while maintaining the structure of the residual block.

**Aligning Clustering Matrices.** To simplify the folding process, we assert that $\mathbf{U}_{l-1} = \mathbf{U}_l^{(2)}$. This ensures consistency in the clustering across the residual block, reducing the need for additional transformations between layers. As a result, the folding costs for the preceding layer and the current layer can be summed directly:

$$J_{\text{tot}} = J_l^{(2)} + J_{l-1}.$$

**Total Approximation Error.** The total approximation error for folding the residual block is defined as:

$$J_{\text{tot}} = \|\mathbf{W}_{\text{tot}} - \mathbf{C}_l^{(2)}\mathbf{W}_{\text{tot}}\|_F^2,$$

where:

$$\mathbf{W}_{\text{tot}} = \begin{bmatrix} \mathbf{W}_{l-1} & \mathbf{W}_l^{(2)} \end{bmatrix}.$$

Here, $\mathbf{W}_{\text{tot}}$ combines the weights of both layers in the residual block into a single representation. This unified view allows the clustering process to be applied holistically, ensuring that redundancies across the entire block are captured and reduced.

By asserting $\mathbf{U}_{l-1} = \mathbf{U}_l^{(2)}$ and summing the individual folding costs $J_l^{(2)}$ and $J_{l-1}$, we achieve a compact representation of the residual block with minimal approximation error. This approach ensures that the compressed residual block remains effective while reducing redundancy in the weights.

## F.2 Residual Blocks with Non-Identity Shortcuts

Consider a Residual Block with a shortcut represented by a weight matrix $\mathbf{W}_{l,s}$, and a preceding layer decomposed using a clustering matrix $\mathbf{U}_{l-1}$. The projection matrix is defined as:

$$\mathbf{C}_{l-1} = \mathbf{U}_{l-1}\left(\mathbf{U}_{l-1}^T\mathbf{U}_{l-1}\right)^{-1}\mathbf{U}_{l-1}^T.$$

This decomposition allows for approximating and clustering the preceding layer's weights while maintaining their representational capacity. The corresponding approximation for the residual block satisfies:

$$\mathbf{y}_l \approx \sigma\left(\mathbf{W}_l^{(2)}\sigma\left(\mathbf{W}_l^{(1)}\mathbf{C}_{l-1}^T\mathbf{x}_{l-1}\right) + \mathbf{W}_{l,s}\mathbf{C}_{l-1}^T\mathbf{x}_{l-1}\right),$$

where:

- $\mathbf{W}_l^{(2)}$ is the weight matrix of the second layer in the residual block,
- $\mathbf{W}_l^{(1)}$ is the weight matrix of the first layer in the residual block,
- $\mathbf{W}_{l,s}$ is the shortcut connection weight matrix,
- $\sigma(\cdot)$ represents the activation function.

**Decomposition of Weight Matrices.** The weights $\mathbf{W}_l^{(2)}$ and $\mathbf{W}_{l,s}$ are decomposed using their respective clustering matrices. For $\mathbf{W}_l^{(2)}$, the decomposition is:

$$\mathbf{C}_l^{(2)} = \mathbf{U}_l^{(2)} \left( \mathbf{U}_l^{(2)T} \mathbf{U}_l^{(2)} \right)^{-1} \mathbf{U}_l^{(2)T}.$$

For $\mathbf{W}_{l,s}$, the decomposition is:

$$\mathbf{C}_{l,s} = \mathbf{U}_{l,s} \left( \mathbf{U}_{l,s}^T \mathbf{U}_{l,s} \right)^{-1} \mathbf{U}_{l,s}^T.$$

Substituting these decompositions into the approximation yields:

$$\mathbf{y}_l \approx \sigma \left( \mathbf{C}_l^{(2)} \mathbf{U}_l^{(2)T} \mathbf{W}_l^{(2)} \sigma \left( \mathbf{W}_l^{(1)} \mathbf{C}_{l-1}^T \mathbf{x}_{l-1} \right) + \mathbf{C}_{l,s} \mathbf{W}_{l,s} \mathbf{C}_{l-1}^T \mathbf{x}_{l-1} \right).$$

**Consistency Constraint and Total Approximation Error.** To simplify the folding process and ensure consistency across the layers, we introduce the constraint:

$$\mathbf{U}_{l,s} = \mathbf{U}_l^{(2)}.$$

This ensures that the same clustering matrix is used for both the shortcut weights $\mathbf{W}_{l,s}$ and the second layer's weights $\mathbf{W}_l^{(2)}$. By adding the individual folding costs $J_l^{(2)}$ and $J_{l,s}$, we ensure that Lemma 1 holds, leading to the total approximation error for the residual block:

$$J_{\text{tot}} = J_l^{(2)} + J_{l,s}.$$

**Unified Approximation for Residual Blocks.** The total approximation error can be expressed compactly as:

$$J_{\text{tot}} = \|\mathbf{W}_{\text{tot}} - \mathbf{C}_l^{(2)} \mathbf{W}_{\text{tot}}\|_F^2,$$

where:

$$\mathbf{W}_{\text{tot}} = \left[ \mathbf{W}_{l,s} \mid \mathbf{W}_l^{(2)} \right].$$

Here, $\mathbf{W}_{\text{tot}}$ combines the shortcut weights $\mathbf{W}_{l,s}$ and the second-layer weights $\mathbf{W}_l^{(2)}$ into a single matrix. This unified representation allows the folding process to be applied holistically, reducing redundancies across the entire residual block.

The decomposition of weights in residual blocks with non-identity shortcuts introduces a consistent clustering mechanism for both the shortcut and the second layer. By ensuring that $\mathbf{U}_{l,s} = \mathbf{U}_l^{(2)}$, we maintain alignment in the clustering process, leading to a compact and efficient representation with minimal approximation error.

# G  HANDLING BATCH NORMALIZATION LAYERS

Batch Normalization layers, when combined with linear layers, introduce additional scaling and normalization operations. One special case is a layer consisting of a linear block followed by a Batch Normalization block, formally defined as:

$$\mathbf{z}_{l+1} = \mathbf{W}_{l+1} \sigma(\boldsymbol{\Sigma}_s \boldsymbol{\Sigma}_n \mathbf{W}_l \mathbf{x}_{l-1}),$$

where:

- $\mathbf{W}_l$: weight matrix of the linear block,
- $\boldsymbol{\Sigma}_s$: Batch Normalization scaling matrix,
- $\boldsymbol{\Sigma}_n$: Batch Normalization normalization matrix,
- $\mathbf{W}_{l+1}$: weight matrix of the subsequent layer,
- $\sigma(\cdot)$: activation function applied element-wise.

A design choice in handling such layers is to decompose $\mathbf{\Sigma}_s$, $\mathbf{\Sigma}_n$, and $\mathbf{W}_l$ separately while preserving the original structure of the layer. This ensures that the scaling, normalization, and linear blocks are treated as distinct functional units. The decomposed approximation for the layer can then be expressed as:

$$\mathbf{z}_{l+1} \approx \tilde{\mathbf{z}}_{l+1} = \mathbf{W}_{l+1}\mathbf{C}_s^T \sigma(\mathbf{C}_s\mathbf{\Sigma}_s\mathbf{C}_n\mathbf{\Sigma}_n\mathbf{C}_l\mathbf{W}_l\mathbf{x}_{l-1}),$$

where the projection matrices $\mathbf{C}_s$, $\mathbf{C}_n$, and $\mathbf{C}_l$ are defined as:

$$\mathbf{C}_s = \mathbf{U}_s(\mathbf{U}_s^T\mathbf{U}_s)^{-1}\mathbf{U}_s^T = \mathbf{U}_s\mathbf{M}_s,$$
$$\mathbf{C}_n = \mathbf{U}_n(\mathbf{U}_n^T\mathbf{U}_n)^{-1}\mathbf{U}_n^T = \mathbf{U}_n\mathbf{M}_n,$$
$$\mathbf{C}_l = \mathbf{U}_l(\mathbf{U}_l^T\mathbf{U}_l)^{-1}\mathbf{U}_l^T = \mathbf{U}_l\mathbf{M}_l.$$

Here, $\mathbf{U}_s$, $\mathbf{U}_n$, and $\mathbf{U}_l$ are clustering matrices, and $\mathbf{M}_s$, $\mathbf{M}_n$, and $\mathbf{M}_l$ are normalization terms.

**Clustering Assumptions.** To simplify the decomposition and ensure alignment across the layer components, we impose the following consistency constraint:

$$\mathbf{U}_s = \mathbf{U}_n = \mathbf{U}_l.$$

This assumption ensures that the same clustering structure is applied to the scaling, normalization, and linear blocks, leading to a unified decomposition. Under this assumption, the approximation becomes:

$$\tilde{\mathbf{z}}_{l+1} = \mathbf{W}_{l+1}\mathbf{C}_l^T \sigma(\mathbf{U}_l\mathbf{M}_l\mathbf{W}_{b,l}\mathbf{U}_l\mathbf{M}_l\mathbf{\Sigma}_n\mathbf{U}_l\mathbf{M}_l\mathbf{W}_l\mathbf{x}_{l-1}),$$

where $\mathbf{W}_{b,l}$ represents the intermediate scaling factors.

**Applying Diagonal Properties.** Using Lemma 3, we observe that the normalization and scaling matrices can be represented as diagonal matrices:

$$\tilde{\mathbf{z}}_{l+1} = \mathbf{W}_{l+1}\mathbf{C}_l^T \sigma(\mathbf{U}_l\text{Diag}(\mathbf{M}_l\text{diag}(\mathbf{W}_{b,l}))\text{Diag}(\mathbf{M}_l\text{diag}(\mathbf{\Sigma}_n))\mathbf{M}_l\mathbf{W}_l\mathbf{x}_{l-1}).$$

Furthermore, by applying Lemma 4, we rewrite this expression as:

$$\tilde{\mathbf{z}}_{l+1} = \mathbf{W}_{l+1}\mathbf{C}_l^T \sigma(\text{Diag}(\mathbf{C}_l\text{diag}(\mathbf{W}_{b,l}))\text{Diag}(\mathbf{C}_l\text{diag}(\mathbf{\Sigma}_n))\mathbf{C}_l\mathbf{W}_l\mathbf{x}_{l-1}).$$

This shows that the diagonal structure of the scaling and alignment matrices is preserved through the decomposition, maintaining the original behavior of the Batch Normalization block.

**Compression Cost.** According to the definition of the Model Folding problem and using the properties stated in Lemma 5, the compression cost for the layer can be expressed as:

$$J_{tot} = \|\mathbf{W}_{tot} - \mathbf{C}_l\mathbf{W}_{tot}\|_F^2,$$

where:

$$\mathbf{W}_{tot} = \begin{bmatrix} \mathbf{W}_{l+1}^T & \mathbf{W}_l & \text{diag}(\mathbf{\Sigma}_s) & \text{diag}(\mathbf{\Sigma}_n) \end{bmatrix}.$$

This cost quantifies the approximation error introduced by clustering the weights, scaling, and normalization matrices while preserving the layer's functional structure.

By decomposing the Batch Normalization and linear blocks separately and aligning their clustering structures ($\mathbf{U}_s = \mathbf{U}_n = \mathbf{U}_l$), we ensure that the original diagonal properties of the scaling and normalization matrices are preserved. The resulting compression cost captures the overall error of folding the entire layer into a compact representation.

## G.1 ALGORITHMIC DESCRIPTION OF FOLD-AR

The Fold-AR algorithm for a single layer combines the Batch Normalization components and layer weights into a compact representation, followed by clustering to reduce redundancy. The steps are described in Algorithm 1.

---

**Algorithm 1** Fold-AR for a Single Layer

---

**Require:** $\boldsymbol{\Sigma}_s, \boldsymbol{\Sigma}_n, \mathbf{W}_l, \mathbf{W}_{l+1}$ ▷ Input components of the layer
 1: Compute the normalized weight matrix: $\hat{\mathbf{W}}_l \leftarrow \boldsymbol{\Sigma}_n \mathbf{W}_l$
 2: Construct the combined weight matrix: $\mathbf{W}_{\text{tot}} \leftarrow \begin{bmatrix} \mathbf{W}_{l+1}^T & \hat{\mathbf{W}}_l & \text{diag}(\boldsymbol{\Sigma}_s) \end{bmatrix}$
 3: Solve the clustering problem:

$$\mathbf{U} \leftarrow \arg\min_{\mathbf{U}} \|\mathbf{W}_{\text{tot}} - \mathbf{U}(\mathbf{U}^T\mathbf{U})^{-1}\mathbf{U}^T\mathbf{W}_{\text{tot}}\|_F^2$$

subject to $\mathbf{U}^T \in \{0,1\}^{m \times n}$ and $m < n$
 4: Update the scaling matrix: $\boldsymbol{\Sigma}_s \leftarrow (\mathbf{U}^T\mathbf{U})^{-1}\mathbf{U}^T\boldsymbol{\Sigma}_s\mathbf{U}$
 5: Update the second-layer weights: $\mathbf{W}_{l+1}^T \leftarrow \mathbf{U}^T\mathbf{W}_{l+1}^T$
 6: Update the current-layer weights: $\hat{\mathbf{W}}_l \leftarrow (\mathbf{U}^T\mathbf{U})^{-1}\mathbf{U}^T\hat{\mathbf{W}}_l$
 7: **for** $c = 1, \ldots, m$ **do** ▷ Adjust scaling factors for each cluster
 8:     Compute cluster size: $N_c \leftarrow \sum_i \mathbb{I}(\mathbf{U}_{i,c} = 1)$ ▷ $\mathbb{I}(\cdot)$ is the indicator function
 9:     Compute intra-cluster correlation:

$$E[c] \leftarrow \frac{1}{N_c^2 - N_c} \sum_{i,j} \frac{\hat{\mathbf{w}}_{l,i,:} \cdot \hat{\mathbf{w}}_{l,j,:}^T}{\sqrt{\|\hat{\mathbf{w}}_{l,i,:}\|^2 \|\hat{\mathbf{w}}_{l,j,:}\|^2}} \mathbb{I}(\mathbf{U}_{i,c} = \mathbf{U}_{j,c} = 1)\mathbb{I}(i \neq j)$$

10:     Update the scaling factor for cluster $c$:

$$(\boldsymbol{\Sigma}_s)_{c,c} \leftarrow (\boldsymbol{\Sigma}_s)_{c,c} \frac{N_c}{\sqrt{N_c + (N_c^2 - N_c)E[c]}}$$

11: **end for**

---

EXPLANATION OF KEY STEPS

**1. Combining Normalization and Weights.** The normalization matrix $\boldsymbol{\Sigma}_n$ is diagonal, and multiplying it with the weight matrix $\mathbf{W}_l$ produces the normalized weight matrix:

$$\hat{\mathbf{W}}_l = \boldsymbol{\Sigma}_n \mathbf{W}_l.$$

This step integrates the normalization operation into the weights of the current layer, reducing the complexity of subsequent computations.

**2. Construction of Combined Weight Matrix.** The combined matrix $\mathbf{W}_{\text{tot}}$ is defined as:

$$\mathbf{W}_{\text{tot}} = \begin{bmatrix} \mathbf{W}_{l+1}^T & \hat{\mathbf{W}}_l & \text{diag}(\boldsymbol{\Sigma}_s) \end{bmatrix}.$$

This matrix aggregates the second-layer weights ($\mathbf{W}_{l+1}^T$), the normalized current-layer weights ($\hat{\mathbf{W}}_l$), and the scaling factors ($\text{diag}(\boldsymbol{\Sigma}_s)$) into a single representation, preparing them for joint clustering.

**3. Clustering.** The projection matrix $\mathbf{U}$ is computed by solving the clustering problem:

$$\mathbf{U} = \arg\min_{\mathbf{U}} \|\mathbf{W}_{\text{tot}} - \mathbf{U}(\mathbf{U}^T\mathbf{U})^{-1}\mathbf{U}^T\mathbf{W}_{\text{tot}}\|_F^2,$$

subject to $\mathbf{U}^T \in \{0,1\}^{m \times n}$ and $m < n$. The clustering minimizes the reconstruction error by projecting the combined weights into a lower-dimensional space defined by $m$ clusters.

**4. Scaling Adjustments.** To ensure proper scaling within each cluster, the diagonal elements of $\boldsymbol{\Sigma}_s$ are updated. For each cluster $c$, the adjustment considers the size of the cluster ($N_c$) and the intra-cluster correlation ($E[c]$):

$$(\boldsymbol{\Sigma}_s)_{c,c} \leftarrow (\boldsymbol{\Sigma}_s)_{c,c} \frac{N_c}{\sqrt{N_c + (N_c^2 - N_c)E[c]}}.$$

The intra-cluster correlation $E[c]$ is computed as a normalized dot product, capturing the redundancy among the weights within the same cluster. This adjustment preserves the scaling properties of the original layer.

**5. Final Updates.**   The current-layer weights $\hat{\mathbf{W}}_l$ and second-layer weights $\mathbf{W}_{l+1}^T$ are updated to align with the clustered representation:

$$\hat{\mathbf{W}}_l \leftarrow (\mathbf{U}^T\mathbf{U})^{-1}\mathbf{U}^T\hat{\mathbf{W}}_l, \quad \mathbf{W}_{l+1}^T \leftarrow \mathbf{U}^T\mathbf{W}_{l+1}^T.$$

These updates ensure consistency between the clustered weights and the projection matrix $\mathbf{U}$.

This algorithm combines clustering, scaling adjustments, and weight updates to compress the layer while preserving its functional properties. The clustering step minimizes redundancy, and the final updates align all components of the layer with the clustered structure.

## H   FOLDING SIMILAR CHANNELS IN MLPS

For fully connected networks, where two successive layers are defined as:

$$\mathbf{x}_l = \sigma(\mathbf{W}_l\mathbf{x}_{l-1}) \text{ and } \mathbf{x}_{l+1} = \sigma(\mathbf{W}_{l+1}\mathbf{x}_l),$$

where $\mathbf{x}_l$ represents the activations of layer $l$, $\mathbf{W}_l$ and $\mathbf{W}_{l+1}$ are the weight matrices, and $\sigma$ is the activation function. The channels of the layer are defined as the coordinates $\mathbf{x}_{l,i}$ of the vector $\mathbf{x}_l$. Each channel corresponds to a specific dimension in the activations.

The folding cost $J_l$ for the $l$-th layer is defined as:

$$J_l = \left\|\mathbf{W}_l - \mathbf{C}_l\mathbf{W}_l\right\|_F^2 + \left\|\mathbf{W}_{l+1}^T - \mathbf{C}_l\mathbf{W}_{l+1}^T\right\|_F^2,$$

where $\mathbf{C}_l$ is a clustering matrix. This cost function represents the optimization objective to minimize the approximation error introduced by folding (clustering) the weights of the $l$-th layer. The first term measures the reconstruction error for the weights $\mathbf{W}_l$, while the second term measures the reconstruction error for the weights $\mathbf{W}_{l+1}$ under the transformation $\mathbf{C}_l$. Together, these terms ensure that the clustering transformation preserves the structure and relationships of the weights across layers.

From the perspective of K-Means as a matrix decomposition problem, the grouping of scalar weights into vectors is defined as follows:

$$\mathbf{W}_l = \begin{bmatrix} \mathbf{p}_1^T \\ \mathbf{p}_2^T \\ \vdots \\ \mathbf{p}_n^T \end{bmatrix} \text{ and } \mathbf{W}_{l+1} = \begin{bmatrix} \mathbf{q}_1 & \mathbf{q}_2 & \cdots & \mathbf{q}_n \end{bmatrix},$$

where $\mathbf{p}_i^T$ are the rows of $\mathbf{W}_l$ and $\mathbf{q}_i$ are the columns of $\mathbf{W}_{l+1}$. These groupings reflect the natural structure of the weight matrices in fully connected layers:

- Each row of $\mathbf{W}_l$ represents the weights associated with a specific output channel of layer $l$.
- Each column of $\mathbf{W}_{l+1}$ represents the weights associated with a specific input channel of layer $l+1$.

In this formulation, the rows $\mathbf{p}_i^T$ and columns $\mathbf{q}_i$ are treated as vectors to be clustered by the matrix $\mathbf{C}_l$, which aligns with the K-Means decomposition perspective. The clustering matrix $\mathbf{C}_l$ maps these weights into representative clusters, preserving the relationships between input and output channels across layers while enabling efficient compression.

## I   FOLDING SIMILAR CHANNELS IN CONVOLUTIONAL LAYERS

For convolutional layers, two successive layers can be defined as:

$$\mathcal{X}_l = \sigma(\mathcal{W}_l * \mathcal{X}_{l-1}) \quad \text{and} \quad \mathcal{X}_{l+1} = \sigma(\mathcal{W}_{l+1} * \mathcal{X}_l),$$

where $\mathcal{X}_l$ is a 3-dimensional feature tensor with values $\mathcal{X}_{c_o,i,j}^{(l)}$. The first dimension, $c_o$, corresponds to the output channels, while $i$ and $j$ represent spatial pixel locations. The 4-dimensional weight tensor $\mathcal{W}_l$ has values $\mathcal{W}_{c_o,c_i,i,j}^{(l)}$, where:

- $c_o$ corresponds to the output channels of $\mathcal{X}_l$,
- $c_i$ corresponds to the input channels of $\mathcal{X}_{l-1}$.

To simplify and compress the network, we decompose the weight tensor $\mathcal{W}_l$ such that output channels of $\mathcal{X}_l$ (i.e., the values $\mathcal{X}^{(l)}_{c_o,i,j}$ for $c_o = 1, \ldots, c_{\text{out}}$), which are similar in some sense, are merged. This folding problem is defined as:

$$J_l = \|\mathcal{W}_l - \mathcal{C}_l \circ \mathcal{W}_l\|_T^2 + \|\mathcal{W}_{l+1} - \mathcal{W}_{l+1} \circ \mathcal{C}_l\|_T^2,$$

where $\mathcal{C}_l$ corresponds to a $1 \times 1$ convolution parameterized by the clustering matrix $\mathbf{C}_l$, with $\mathcal{C}^{(l)}_{c,1,1} = \mathbf{C}_{l,c,c'}$.

From this definition, it follows that:

$$J_l = \|\mathbf{W}_l - \mathbf{C}_l \mathbf{W}_l\|_T^2 + \|\mathbf{W}_{l+1} - \mathbf{W}_{l+1} \mathbf{C}_l^T\|_T^2,$$

where the weight tensors $\mathcal{W}_l$ and $\mathcal{W}_{l+1}$ are mapped to matrices $\mathbf{W}_l$ and $\mathbf{W}_{l+1}$ as follows:

$$\mathbf{W}_l = \begin{bmatrix} \text{vec}(\mathcal{W}^{(l)}_{1,1,:,:})^T & \text{vec}(\mathcal{W}^{(l)}_{1,2,:,:})^T & \cdots & \text{vec}(\mathcal{W}^{(l)}_{1,c_{\text{in}},:,:})^T \\ \text{vec}(\mathcal{W}^{(l)}_{2,1,:,:})^T & \text{vec}(\mathcal{W}^{(l)}_{2,2,:,:})^T & \cdots & \text{vec}(\mathcal{W}^{(l)}_{2,c_{\text{in}},:,:})^T \\ \vdots & \vdots & \ddots & \vdots \\ \text{vec}(\mathcal{W}^{(l)}_{c_{\text{out}},1,:,:})^T & \text{vec}(\mathcal{W}^{(l)}_{c_{\text{out}},2,:,:})^T & \cdots & \text{vec}(\mathcal{W}^{(l)}_{c_{\text{out}},c_{\text{in}},:,:})^T \end{bmatrix}.$$

This means that each convolutional filter contributing to an output channel $c_o$ is flattened and stacked into a vector, forming the $c_o$-th row of the matrix $\mathbf{W}_l$. Similarly, for $\mathcal{W}_{l+1}$, each filter associated with the $c_i$-th input channel is flattened and stacked into a vector, forming a column of the matrix $\mathbf{W}_{l+1}$:

$$\mathbf{W}_{l+1} = \begin{bmatrix} \text{vec}(\mathcal{W}^{(l+1)}_{1,1,:,:}) & \text{vec}(\mathcal{W}^{(l+1)}_{1,2,:,:}) & \cdots & \text{vec}(\mathcal{W}^{(l+1)}_{1,c_{\text{in}},:,:}) \\ \text{vec}(\mathcal{W}^{(l+1)}_{2,1,:,:}) & \text{vec}(\mathcal{W}^{(l+1)}_{2,2,:,:}) & \cdots & \text{vec}(\mathcal{W}^{(l+1)}_{2,c_{\text{in}},:,:}) \\ \vdots & \vdots & \ddots & \vdots \\ \text{vec}(\mathcal{W}^{(l+1)}_{c_{\text{out}},1,:,:}) & \text{vec}(\mathcal{W}^{(l+1)}_{c_{\text{out}},2,:,:}) & \cdots & \text{vec}(\mathcal{W}^{(l+1)}_{c_{\text{out}},c_{\text{in}},:,:}) \end{bmatrix}.$$

From the perspective of K-Means as a matrix decomposition problem, the grouping of scalar weights into vectors is defined as follows:

$$\mathbf{W}_l = \begin{bmatrix} \mathbf{p}_1^T \\ \mathbf{p}_2^T \\ \vdots \\ \mathbf{p}_n^T \end{bmatrix} \quad \text{and} \quad \mathbf{W}_{l+1} = \begin{bmatrix} \mathbf{q}_1 & \mathbf{q}_2 & \cdots & \mathbf{q}_n \end{bmatrix},$$

where:

$$\mathbf{p}_i^T = \begin{bmatrix} \text{vec}(\mathcal{W}^{(l)}_{i,1,:,:})^T & \text{vec}(\mathcal{W}^{(l)}_{i,2,:,:})^T & \cdots & \text{vec}(\mathcal{W}^{(l)}_{i,c_{\text{in}},:,:})^T \end{bmatrix},$$

and:

$$\mathbf{q}_j = \begin{bmatrix} \text{vec}(\mathcal{W}^{(l+1)}_{1,j,:,:})^T & \text{vec}(\mathcal{W}^{(l+1)}_{2,j,:,:})^T & \cdots & \text{vec}(\mathcal{W}^{(l+1)}_{c_{\text{out}},j,:,:})^T \end{bmatrix}^T.$$

In this formulation, the rows $\mathbf{p}_i^T$ of $\mathbf{W}_l$ and columns $\mathbf{q}_j$ of $\mathbf{W}_{l+1}$ are grouped into clusters for the folding process, aligning with the K-Means decomposition perspective.

## J  FOLDING SIMILAR CHANNELS IN LLAMAMLP AND LLAMAATTENTION

### J.1  FOLDING SIMILAR CHANNELS IN LLAMAMLP

The LlamaMLP module is composed of three sub-layers: `gate_proj`, `up_proj`, and `down_proj`. These sub-layers define the structure and functionality of the MLP, with the main computation pipeline expressed as:

$$\texttt{down\_proj(act\_fn(gate\_proj}(x)) \times \texttt{up\_proj}(x)).$$

We cluster similar channels in both the output channel and input channel of each sub-layer.

**Input Channel Folding.** To fold the **input channels** of LlamaMLP, we simultaneously consider the input dimensions of both `gate_proj` and `up_proj` layers, as they collectively define the effective input to the `gate_up` sub-layer. The input channels of `gate_proj` and `up_proj` are clustered respectively using methods similar to those applied in standard MLP layers.

**Output Channel Folding.** To fold the **output channels** of LlamaMLP, we first consider the output channels of both `gate_proj` and `up_proj` by clustering and adjusting the input channel of the `down_proj`. Subsequently, we adjust the output channel of `down_proj` according to the residual connection used outside of LlamaMLP.

### J.2 FOLDING SIMILAR CHANNELS IN LLAMAATTENTION

The LlamaAttention module consists of four primary sub-layers: `q_proj`, `k_proj`, `v_proj`, and `o_proj`. These sub-layers define the query, key, value, and output projections, respectively. For clarity and simplicity, we conceptualize `q_proj`, `k_proj`, and `v_proj` as a unified sub-layer referred to as `q_k_v`, which computes the intermediate representations required for attention calculations. The `o_proj` sub-layer processes the final output of the attention mechanism. We treat the attention head as the structure to be folded in LlamaAttention. By reshaping the weights of each sub-layer into an MLP-like tensor, we can cluster similar heads, similar to how it is done for a standard MLP layer.

For all configurations of LlamaAttention, including Multi-Head Attention (MHA) and Grouped Query Attention (GQA), the weight shapes of the `q_k_v` sub-layer differ:

- In MHA, the weights for `q`, `k`, and `v` projections share the same shape: $[\text{num\_heads} \times \text{head\_dim}, \text{hidden\_size}]$.
- In GQA, the weights for `k` and `v` projections have the shape: $[\text{num\_kv\_heads} \times \text{head\_dim}, \text{hidden\_size}]$.

**Output Channel Folding.** When performing **output channel folding** for the LlamaAttention layer, the clustering of the `o_proj` sub-layer's output channels is dictated by the residual connection outside of LlamaAttention, ensuring alignment with the clustering results from previous modules. Specifically:

- The `o_proj` weights, originally shaped as $[\text{num\_heads} \times \text{head\_dim}, \text{hidden\_size}]$, are reshaped into $[\text{num\_heads}, \text{head\_dim}, \text{hidden\_size}]$, clustered along the first dimension (`num_heads`), and then reshaped back to their original form.
- For clustering within the `q_k_v` sub-layer, the weights for `q`, `k`, and `v` are reshaped into $[\text{num\_heads}, \text{head\_dim}, \text{hidden\_size}]$ (or $[\text{num\_kv\_heads}, \text{head\_dim}, \text{hidden\_size}]$ for `k` and `v` in GQA) and clustered along the first dimension (`num_heads` or `num_kv_heads`). After clustering, the weights are reshaped back to their original dimensions.

**Input Channel Folding.** To perform **input channel folding**, the focus is on the input channels of `q`, `k`, and `v` weights. Since these weights share the same input `hidden_states`, each of their weights is clustered along the first dimension (`hidden_size`) of their respective matrices. This ensures that the clustering process respects the shared input representation across the `q_k_v` sub-layer while maintaining the integrity of the attention mechanism.

## K COMPARISON WITH KNOWLEDGE DISTILLATION

We evaluated some data-free knowledge distillation (KD) methods (Micaelli & Storkey, 2019; Chen et al., 2019; Fang et al., 2020; Yu et al., 2023), on an NVIDIA A100 GPU, for all methods using the same pre-trained teacher model, data loader, and student model setup for consistency. The full model is a ResNet18 pre-defined by torchvision and trained on CIFAR10, while the student models for each KD method share the same architecture but differ in the number of channels across all layers to achieve the desired sparsity levels. Specifically, in ResNet18, the number of output channels for all blocks is a multiple of 64, which is also the number of output channels in the first convolutional layer. To reduce the model's channel dimensions, we scale this base hyperparameter by a reduction factor, effectively reducing the width of all layers proportionally. The following table presents the test

accuracy of compressed by KD methods and model folding on CIFAR10 test dataset.The time taken to achieve each accuracy is provided in parentheses next to the corresponding accuracy value. From the table, it is evident that the proposed model folding achieves model compression within seconds, even at high sparsity levels, compared to other KD methods that require tens of hours to complete.

| Sparsity | Full model | 10% | 25% | 50% | 70% |
|---|---|---|---|---|---|
| ABM (Micaelli & Storkey, 2019) | 94.72 | 93.30 (17h19m) | 91.99 (16h8m) | 89.42 (15h30m) | 85.43 (13h23m) |
| DFAD (Chen et al., 2019) | 94.72 | 93.79 (2h31m) | 93.52 (2h3m) | 92.04 (2h1m) | 89.67 (1h54m) |
| DAFL (Fang et al., 2020) | 94.72 | 71.73 (16h48m) | 77.80 (15h39m) | 68.06 (15h19m) | 53.86(76h34m) |
| SpaceshipNet (Yu et al., 2023) | 94.72 | 94.50 (42h33m) | 93.95 (40h3m) | 92.96 (37h57m) | 91.53 (27h10m) |
| **Model Folding (ours)** | 94.72 | 94 (56.35s) | 92 (53.55s) | 88 (55.75s) | 82 (54.95s) |

Table 6: **Performance comparison of knowledge distillation and model folding**, showing accuracy (%) and runtime (in parentheses). The sparsity levels indicate the percentage of weights pruned.

# L    INFERENCE SPEED OF FOLDED MODELS ON EDGE DEVICES

We apply model folding on a LeNet5 model pre-trained on FashionMNIST with different sparsity, and then evaluate the folded models on NVIDIA Jetson Nano, ESP-EYE, and Arduino Nano 33 BLE. All models are converted and executed as a float32 Tensorflow Lite model in all devices.

| Sparsity | 10% | | | 25% | | | 50% | | | 70% | | |
|---|---|---|---|---|---|---|---|---|---|---|---|---|
| | Runtime | RAM | Flash | Runtime | RAM | Flash | Runtime | RAM | Flash | Runtime | RAM | Flash |
| NVIDIA Jetson Nano (NVIDIA, 2024) | 2ms | 59.5K | 3.4M | 2ms | 55.7K | 2.8M | 1ms | 48.0K | 1.9M | 1ms | 36.5K | 1.2M |
| ESP-EYE (Espressif Systems, 2024) | 2591ms | 59.5K | 3.4M | 1868ms | 55.7K | 2.8M | 1532ms | 48.0K | 1.9M | 1186ms | 36.5K | 1.2M |
| Arduino Nano 33 BLE Sense (Arduino, 2024) | 6831ms | 59.5K | 3.4M | 3726ms | 55.7K | 2.8M | 4218ms | 48.0K | 1.9M | 2969ms | 36.5K | 1.2M |

Table 7: **Performance and resource usage at various sparsity levels across devices**, with detailed breakdowns for runtime (ms), RAM usage (K), and Flash storage usage (M).

# M    DEEP INVERSION SAMPLE IMAGES

Deep Inversion (DI) (Yin et al., 2020) generates synthetic images from the uncompressed network by optimizing noise to match the internal statistics stored in BatchNorm layers. These images, exemplified in Fig. 13, which reflect the original data's statistical properties, are used during model folding to restore data statistics in the compressed network, ensuring accuracy without requiring external data.

# N    FURTHER RELATED WORK

Model folding intersects with several established approaches in model compression, network architecture optimization and model merging. This section outlines key related works that inspired the development of model folding, highlighting both their contributions and limitations.

## N.1    MODEL COMPRESSION

Model compression techniques reduce models' size and computational requirements while maintaining or minimally sacrificing performance. Various methods have been developed. Most can be classified as pruning, quantization, knowledge distillation, and low-rank factorization. Traditional pruning techniques (Han et al., 2015; LeCun et al., 1989; Li et al., 2016b; Hassibi et al., 1993; Entezari & Saukh, 2020), structured or unstructured, involve removing weights, neurons, or filters that are deemed less important, typically measured by the magnitude of their contributions (*e.g.*,, $L_1$ or $L_2$ norm) (Entezari & Saukh, 2020; Li et al., 2017; Cheng et al., 2023). While effective in reducing the size of the model, pruning often leads to a degradation of performance that requires fine-tuning or complete retraining of the network (Cheng et al., 2023; Han et al., 2015; Frankle & Carbin, 2018; Frantar & Alistarh, 2022; He et al., 2018). Quantization (Gupta et al., 2015; Zhou et al., 2017; Li

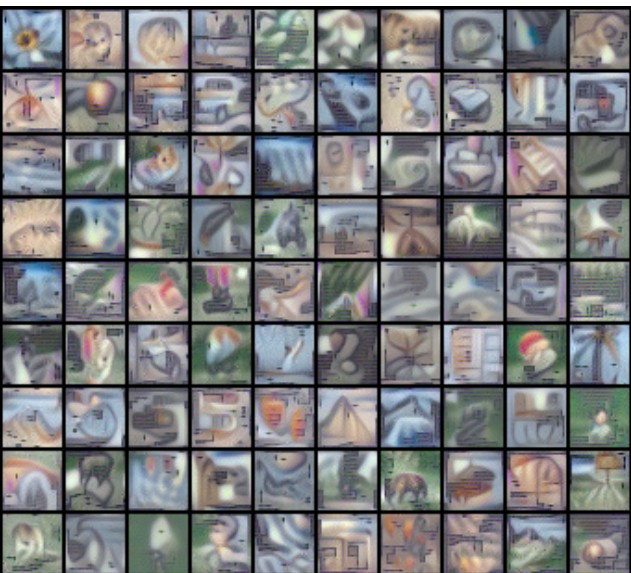

Figure 13: **Sample images generated by Deep Inversion (Yin et al., 2020) using ResNet18 trained on CIFAR100.** These images are generated from the uncompressed network and used in model folding to restore data statistics in the compressed network.

et al., 2016a) reduces the precision of the numerical values in a model, from floating-point to lower-bit representations (*e.g.*,, 8-bit integers). This approach significantly reduces the model's memory footprint and speeds up computation, especially when combined with hardware accelerators designed for low-precision arithmetic (Gholami et al., 2021). Like pruning, post-training quantization may also require fine-tuning to restore model performance. Knowledge distillation (Hinton et al., 2015) trains a smaller model, called the student, to replicate a well-trained larger model, called the teacher, by mimicking the output of the teacher model, which transfers knowledge between the teacher model and the student model. While effective in transferring knowledge and reducing model size, even approaches that eliminate data dependency using synthetic samples or adversarial distillation (Micaelli & Storkey, 2019; Chen et al., 2019; Fang et al., 2020; Yu et al., 2023; Haroush et al., 2020), the training process for knowledge distillation can be computationally expensive and time-consuming (Hinton et al., 2015; Gou et al., 2021; Martinez et al., 2021). Moreover, knowledge distillation often assumes substantial differences between student and teacher model architectures (Gou et al., 2021). Low-rank factorization decomposes weight matrices into lower-rank matrices to reduce parameter size through such as singular value decomposition (Ren & Zhu, 2023; Horvath et al., 2024) or tensor decomposition (Lebedev et al., 2015; Kim et al., 2016). Approaches such as mixture of experts (Jacobs et al., 1991; Shazeer et al., 2017), subspace-configurable networks (Wang et al., 2024; Papst et al., 2024) and resource-efficient deep subnetworks (Corti et al., 2024b;a), explore dynamic model reconfiguration to minimize the number of active weights during inference.

**Structured pruning**. Structured pruning is of particular interest because it removes entire structures (such as neurons, channels, or layers) (Entezari & Saukh, 2020; Li et al., 2016b; Luo et al., 2017a; Hu et al., 2016; Wen et al., 2016) rather than individual parameters, reducing model complexity while maintaining or even improving performance. This method is especially valuable for enhancing efficiency with easily implemented acceleration in resource-constrained environments (Wang et al., 2020; Liu et al., 2024). However, structured pruning typically requires additional retraining or fine-tuning (He et al., 2017; Liu et al., 2024; Luo et al., 2017b). Recent work by Theus et al. (2024) combines model pruning and fusion using Optimal Transport theory, demonstrating that a significant portion of pruning accuracy can be recovered without access to training data. However, the impact of pruning on the model's data statistics and how to recover them is not addressed.

## N.2 MODEL MERGING

Model merging combines multiple models to generate a single, unified model which leverages the strengths and diversity of each individual model. It particularly benefits ensemble learning and distributed training scenarios, where models are trained independently on different subsets of data or across different devices. Merging can be achieved by averaging the parameters of model trained independently. Recently, multiple methods have been developed to enhance model performance and robustness. MTZ (He et al., 2018) and ZipIt! (Stoica et al., 2024) compress multiple models pre-trained for different tasks by merging them through neuron sharing. Model soup (Wortsman et al., 2022) averages the weights of multiple fine-tuned models from same initialization to improve accuracy and robustness without increasing inference time. Taking permutation invariance of neural networks into account, a finding (Entezari et al., 2022) shows the interpolation between models trained with SGD has no barrier. Git Re-Basin (Ainsworth et al., 2023) utilizes activation matching and weight matching to achieve permutated alignment between models trained from different initialization. REPAIR (Jordan et al., 2022) mitigate variance collapse problem while aligning neurons by rescaling the preactivations of fused models. PAPA leverages a population of diverse models trained on different data variations and slowly pushes the weights of the networks towards the population average (Jolicoeur-Martineau et al., 2024). A recent work (Yamada et al., 2023) shows that for model merging on different datasets, using original or condensed datasets during the model merging process can significantly improve accuracy. However, those methods do not consider model efficiency and internal parameter redundancy. Another recent work (Theus et al., 2024) achieves intra-layer model fusion by integrating optimal transport (Monge, 1781; Kantorovich, 2006; Singh & Jaggi, 2020) to fuse computational structures in the model without fine-tuning. We note that this approach is orthogonal to the problem solved in this paper, as we do not consider intra-layer dependencies.

**Merging multiple computational units**. Merging computational units has been extensively explored in ensemble methods. Wortsman et al. (2022) demonstrate that combining multiple models fine-tuned from the same pretrained initialization enhances both accuracy and robustness. Ainsworth et al. (2023) extend this approach to models trained on the same data with different initializations, albeit with some accuracy loss. Jordan et al. (2022) improve upon Git Re-Basin by adjusting batch normalization layers where applicable. IFM Chen et al. (2023) and ZipIt! Stoica et al. (2024) focus on merging multiple computational units within a single model, pioneering this approach.

