# OpenReview forum: "Forget the Data and Fine-Tuning! Just Fold the Network to Compress"
_ICLR.cc/2025/Conference — ICLR 2025 Poster_

### Official Review · Reviewer_AJd2 · 2024-10-30

**Soundness:** 3
**Presentation:** 3
**Contribution:** 2
**Rating:** 6
**Confidence:** 4

**Summary:**

This paper introduces a method to merge the columns in the weight matrix in a data-free manner. The method is composed of two parts: first, using the clustering method to find and merge the weight matrix. Then, based on the observation that the variance ratio after compressing the model would collapse or overshooting, the authors propose the data-free Fold-AR and Fold-DIR to inverse the model and repair the model. Experiments show that the proposed method can achieve better results than previous data-free methods, especially at high sparsity ratio.

**Strengths:**

1. Rigorous theoretical analysis of the proposed method
2. A comprehensive framework for merging and repairing the model under the data-free setting, also without fine-tuning.
3. The writing of the paper is good

**Weaknesses:**

1. There are plenty of works [1,2,3,4] about how to compress the model under the data-free setting, including the DeepInversion, which is mentioned as part of the method in this paper. These papers are all not mentioned and compared in the paper.
2. The experimental results show that the authors’ proposed methods experience significant performance drops compared to finetuning-based methods. Given that the cost of finetuning remains acceptable but the performance drop is not neglectable, it’s unclear why a tuning-free setting is necessary (since we can get the synthetic by methods like DeepInversion).
3. For methods on LLaMA, [5] also propose a model merging method to compress the model without fine-tuning. Please compare to this paper.

[1] Zero-shot knowledge transfer via adversarial belief matching.
[2] Data-free learning of student networks.
[3] Data-free adversarial distillation.
[4] Data-free knowledge distillation via feature exchange and activation region constraint.
[5] Shortgpt: Layers in large language models are more redundant than you expect.

**Questions:**

Please refer to the weaknesses.

---

> ### Author Response · Authors · 2024-11-22
> **Rebuttal to Reviewer AJd2 (1/2)**
>
> We appreciate the reviewer’s feedback and thoughtful comments on our submission, particularly for recognizing the strength of the theoretical aspect of our analysis.
>
> ## W.1: Comparison with data-free knowledge distillation methods
>
> We evaluated some data-free knowledge distillation (KD) methods [1, 2, 3, 4] on an NVIDIA A100 GPU, for all methods using the same pre-trained teacher model, data loader, and student model setup for consistency. The full model is a ResNet18 pre-defined by torchvision and trained on CIFAR10, while the student models for each KD method share the same architecture but differ in the number of channels across all layers to achieve the desired sparsity levels. Specifically, in ResNet18, the number of output channels for all blocks is a multiple of 64, which is also the number of output channels in the first convolutional layer. To reduce the model's channel dimensions, we scale this base hyperparameter by a reduction factor, effectively reducing the width of all layers proportionally. The following table presents the test accuracy of compressed by KD methods and model folding on CIFAR10 test dataset.The time taken to achieve each accuracy is provided in parentheses next to the corresponding accuracy value. From the table, it is evident that the proposed model folding achieves model compression within seconds, even at high sparsity levels, compared to other KD methods that require tens of hours to complete. We updated the paper in the appendix K, with yellow background text.
>
> | Sparsity                     | Full model | 10%             | 25%             | 50%             | 70%             |
> | ---------------------------- | ---------- | --------------- | --------------- | --------------- | --------------- |
> | ZeroShotKnowledgeTransfer[1] | 94.72      | 93.30% (17h19m) | 91.99% (16h8m)  | 89.42% (15h30m) | 85.43% (13h23m) |
> | DFAD[2]                      | 94.72      | 93.79% (2h31m)  | 93.52% (2h3m)   | 92.04% (2h1m)   | 89.67% (1h54m)  |
> | DAFL[3]                      | 94.72      | 71.73% (16h48m) | 77.80% (15h39m) | 68.06% (15h19m) | 53.86%(76h34m)               |
> | SpaceshipNet[4]              | 94.72      | 94.50% (42h33m) | 93.95% (40h3m)  | 92.96% (37h57m) | 91.53% (27h10m) |
> | Model Folding**(ours)**      | 94.72      | 94% (56.35s)    | 92% (53.55s)    | 88% (55.75s)    | 82% (54.95s)    |
>
> [1] Micaelli, Paul, and Amos Storkey. "Zero-shot Knowledge Transfer via Adversarial Belief Matching." arXiv preprint arXiv:1905.09768 (2019).
>
> [2] Chen, Hanting, et al. "Data-Free Learning of Student Networks." arXiv preprint arXiv:1904.01186 (2019).
>
> [3] Fang, Gongfan, et al. "Data-free adversarial distillation." arXiv preprint arXiv:1912.11006 (2019).
>
> [4] Yu, Shikang, et al. "Data-free knowledge distillation via feature exchange and activation region constraint." Proceedings of the IEEE/CVF Conference on Computer Vision and Pattern Recognition. 2023.
>
>
> ## W.2: Motivation for fine-tuning-free methods
>
> Fine-tuning large pre-trained models, while effective in certain scenarios, is resource-intensive and time-consuming, particularly on low-resource devices such as mobile or edge platforms. These devices often lack the GPU memory and computational capacity required for fine-tuning, making it infeasible in practice. Furthermore, scenarios that demand real-time or on-the-fly distribution – such as personalized recommendations, rapid task-specific model loading, or  serverless model sharing – cannot accommodate the latency introduced by fine-tuning.
>
> In real-world applications, distributing models to a diverse range of heterogeneous user devices (e.g., smartphones with varying hardware capabilities, edge devices in general) presents additional challenges. Fine-tuning on such resource-constrained devices not only becomes computationally prohibitive but also increases energy consumption and risks degrading user experience.
>
> While fine-tuning may remain viable in well-equipped settings, it relies on access to real or synthetic data. Synthetic data generation processes like DeepInversion, although powerful, introduce significant computational overhead, further limiting their practicality for on-device or rapid deployment scenarios.
>
> In contrast, our proposed method, Fold-AR, provides a tuning-free alternative that operates without relying on real or synthetic data. This eliminates the dependency on data availability and computationally expensive generative processes. Our results demonstrate that Fold-AR achieves competitive performance, particularly at high sparsity levels, and outperforms state-of-the-art methods like INN and IFM in these regimes. By addressing the constraints of resource-limited environments, Fold-AR bridges a critical gap, enabling effective compression without the burdens of fine-tuning.

---

> > ### Author Response · Authors · 2024-11-22
> > **Rebuttal to Reviewer AJd2 (2/2)**
> >
> > ## W.3: Comparison with SliceGPT and ShortGPT on LLaMA-7B without fine-tuning
> >
> > We thank the reviewer for highlighting the gap between our data-free, fine-tuning-free method and other data-driven approaches on LLaMA-7B. In response, we have compared two newly released model compression methods, ShortGPT and SliceGPT, applied to LLaMA-7B without fine-tuning. The updated results are presented in the table below. Notably, Model Folding achieves comparable perplexity (PPL) to the data-driven pruning method ShortGPT on WikiText2. The updated details are provided in Table 1 of the revised PDF, with changes highlighted in yellow.
> >
> > | Prune ratio | Method           | Data usage  | WikiText2 ↓ | BoolQ | WinoGrande | ARC-e | ARC-c | Average ↑ |
> > | ----------- | ---------------- | ----------- | ----------- | ----- | ---------- | ----- | ----- | --------- |
> > | **0%**      | LLaMA-7B[1] | /           | 5.68        | 75.05 | 69.93      | 75.34 | 41.89 | 65.55     |
> > | **20%**     | Magnitude Prune  | /           | 36136       | 43.21 | 49.40      | 27.23 | 21.59 | 35.36     |
> > | **20%**     | LLM-Pruner[2]       | Gradients   | 10.53       | 59.39 | 61.33      | 59.18 | 37.18 | 54.27     |
> > | **20%**     | FLAP[3]             | Calibration | 6.87        | 69.63 | 68.35      | 69.91 | 39.25 | 61.79     |
> > | **20%**     | Wanda_sp[4]         | Calibration | 8.22        | 71.25 | 67.09      | 71.09 | 42.58 | 63.00     |
> > | **20%**     | SliceGPT[5]         | Calibration | 7.00        | 57.80 | 67.96      | 62.67 | 36.01 | 56.11     |
> > | **20%**     | ShortGPT[6]         | Calibration | 15.48       | 62.17 | 67.40      | 58.88 | 31.91 | 55.09     |
> > | **20%**     | Model Folding    | /           | 13.33       | 62.29 | 62.19      | 49.83 | 26.37 | 50.17     |
> >
> > [1] Touvron, Hugo, et al. "Llama: Open and efficient foundation language models." arXiv preprint arXiv:2302.13971 (2023).
> >
> > [2] Ma, Xinyin, Gongfan Fang, and Xinchao Wang. "LLM-pruner: on the structural pruning of large language models." Proceedings of the 37th International Conference on Neural Information Processing Systems. 2023.
> >
> > [3] An, Yongqi, et al. "Fluctuation-based Adaptive Structured Pruning for Large Language Models." arXiv preprint arXiv:2312.11983 (2023).
> >
> > [4] Sun, Mingjie, et al. "A simple and effective pruning approach for large language models." arXiv preprint arXiv:2306.11695 (2023).
> >
> > [5] Ashkboos, Saleh, et al. "SliceGPT: Compress large language models by deleting rows and columns." arXiv preprint arXiv:2401.15024 (2024).
> >
> > [6] Men, Xin, et al. "Shortgpt: Layers in large language models are more redundant than you expect." arXiv preprint arXiv:2403.03853 (2024).

---

> > > ### Comment · Reviewer_AJd2 · 2024-11-26
> > >
> > > I appreciate the authors for providing more experimental results and the detailed response. I think the response has solved my concern for the experiments.
> > >
> > > However, I still have some concerns for the motivation (W2). I agree that resources are becoming increasingly constrained for large models. However, methods like those used in current LLM compression techniques typically have low resource requirements. For instance, algorithms like sparseGPT, Wanda, and LLM-Pruner can even operate without GPUs. Given the significant performance degradation in this setting, I still have concerns for the setting of this paper.
> > >
> > > Since that the authors have solved most of my concerns, I increase my score to 6.

---

> > > > ### Author Response · Authors · 2024-11-27
> > > >
> > > > Thank you for your thoughtful feedback and for increasing the score to support our paper. We sincerely appreciate your time and consideration.
> > > >
> > > > We understand your concerns regarding the resource requirements and the setting of our work compared to other methods such as SliceGPT, Wanda, and LLM-Pruner. However, we would like to emphasize that **Model Folding operates in a fundamentally different regime**: it is **fully data-free** and does not rely on fine-tuning, unlike many existing methods that leverage at least a small amount of data or auxiliary computational resources to calibrate the model.
> > > >
> > > > The motivation for Model Folding lies in addressing critical scenarios where access to data is restricted due to sensitivity, privacy, or proprietary concerns. While data-driven LLM compression methods may demonstrate low resource requirements, they still rely on data availability, which our method explicitly avoids.
> > > >
> > > > Moreover, **Model Folding is the first approach, to the best of our knowledge, that achieves performance close to state-of-the-art LLM compression methods in this uniquely constrained, data-free setting**. This highlights its utility in real-world applications where such constraints are non-negotiable, providing an essential alternative to data-driven methods.
> > > >
> > > > We hope this clarifies the distinct motivation and value of Model Folding. Thank you again for your detailed review and for recognizing the strengths of our work.

---

### Official Review · Reviewer_Ao8L · 2024-11-02

**Soundness:** 2
**Presentation:** 3
**Contribution:** 2
**Rating:** 5
**Confidence:** 4

**Summary:**

This paper presents a data-free and fine-tuning-free method for compressing deep neural networks. This is achieved by identifying and merging redundant parameters through a similarity-based approach, followed by data generation with DeepInversion for further repair. The method is flexible and can be applied to both vision models like ResNet by merging channels and LLMs by merging linear weights, demonstrating effectiveness across different architectures. Extensive experiments on ImageNet, CIFAR-10, and CIFAR-100 highlight the method's robustness, with results clearly illustrated through figures.

**Strengths:**

1. Previous methods typically rely on data-driven fine-tuning to restore the performance of compressed models. This paper addresses this limitation by introducing a data-free method that leverages generated data and an efficient activation repair process to recover model accuracy, making it highly practical for real-world applications.
2. The paper includes extensive experiments on both vision and language models, with clear visualizations and figures that offer detailed insights into the method's performance. Notably, at high sparsity levels (>40%), the proposed method achieves significantly better accuracy compared to prior baselines, such as IFM and INN.
3. The presentation is good, with a clear illustration of the entire pipeline in Figure 1. Additionally, the method is straightforward to implement and highly reproducible.

**Weaknesses:**

1. My main concern with this submission is the limited technical novelty. The work largely combines existing methods, including similarity-based model merging for compression [1], model inversion for data generation [2], and REPAIR [3] for statistics alignment. While these components are integrated effectively, the key contributions could be further emphasized to distinguish this work.
2. Although the paper claims the method is data-free, this feature is primarily enabled by DeepInversion rather than a new development within this work. Clarifying the unique aspects of this approach compared to existing data-free methods would strengthen the contribution.
3. The folding results on LLaMA-7B are not particularly competitive, with a Wiki PPL of 13.33, compared to other pruning methods like LLM-Pruner (PPL=10.53) and Wanda (PPL=8.22). The claim that "model folding achieves comparable performance to data-driven methods" may be overstated due to this significant gap in PPL.
4. SliceGPT [4], which similarly merges parameter matrices via a learnable transformation matrix, is another "merging"-based method and should be considered as a relevant baseline for comparison.

Minor Typos: L173 - activation matching matching

[1] Chen, Yiting, Zhanpeng Zhou, and Junchi Yan. "Going Beyond Neural Network Feature Similarity: The Network Feature Complexity and Its Interpretation Using Category Theory." arXiv preprint arXiv:2310.06756 (2023).
[2] Yin, Hongxu, et al. "Dreaming to distill: Data-free knowledge transfer via deepinversion." Proceedings of the IEEE/CVF conference on computer vision and pattern recognition. 2020.
[3] Jordan, Keller, et al. "Repair: Renormalizing permuted activations for interpolation repair." arXiv preprint arXiv:2211.08403 (2022).
[4] Ashkboos, Saleh, et al. "Slicegpt: Compress large language models by deleting rows and columns." arXiv preprint arXiv:2401.15024 (2024).

**Questions:**

Please see weaknesses.

---

> ### Author Response · Authors · 2024-11-22
> **Rebuttal to Reviewer Ao8L (1/2)**
>
> We thank the reviewer's feedback and assessment.
>
>
> ## W.1 and W.2: novelty and the role of data inversion
>
> The problem of data-free model compression without fine-tuning has been addressed in prior works, such as IFM, which use weight matching and aim to maintain batchnorm statistics without data. However, our Model Folding methods (both Fold-AR and Fold-DIR) demonstrate substantial improvements over the state-of-the-art (SoTA), underscoring the challenges of effectively combining existing techniques.
>
> Model Folding’s success relies on two main components: (1) k-Means clustering for structural unit compression: We leverage a factorization-based formulation of k-Means clustering [1], enabling us to theoretically establish its efficiency for model compression. (2) Data-free approximation of post-compression batchnorm statistics: We introduce two approaches: Fold-AR and Fold-DIR. Fold-DIR, which utilizes Deep Inversion, is more resource-intensive, and used to illustrate that different methods can be applied to recover batchnorm statistics without data. An alternative could be to compute the channel covariance matrix at the end of training, which could enhance Fold-AR’s performance at the expense of additional storage. Notably, Fold-AR stands out as a more resource-efficient option while still significantly surpassing existing methods. For example, folding a ResNet18 model pre-trained on CIFAR10 with Fold-AR on an NVIDIA A100 GPU takes an average of 9.55 seconds. This is only one-tenth of Fold-DIR's average runtime of 93.35 seconds, which includes the time required to generate 256 synthetic images (=one batch) and run a forward pass over this batch.
>
> Furthermore, unlike prior SoTA methods [2,3], which primarily exploit weight similarity for compression and limit their evaluation to vision tasks, we extend the scope of Model Folding to LLMs and show decent performance. Importantly, in contrast to all our baselines (with the exception of Magnitude Pruning), our approach requires no data-based calibration during the compression process. Also, since LLaMA-7B does not have any batchnorm layers, neither Fold-AR nor Fold-DIR is applied. So, the strength of the proposed method does not come from Deep Inversion alone.
>
> Thank you for your comment. We updated the introduction of the paper to better articulate our contributions.
>
> [1] Bauckhage, Christian. "K-means clustering is matrix factorization." arXiv preprint arXiv:1512.07548 (2015).
>
> [2] Chen, Yiting, Zhanpeng Zhou, and Junchi Yan. "Going Beyond Neural Network Feature Similarity: The Network Feature Complexity and Its Interpretation Using Category Theory." arXiv preprint arXiv:2310.06756 (2023).
>
> [3] Stoica, George, et al. "Zipit! merging models from different tasks without training." arXiv preprint arXiv:2305.03053 (2023).

---

> > ### Author Response · Authors · 2024-11-22
> > **Rebuttal to Reviewer Ao8L (2/2)**
> >
> > ## W.3 and W.4: Comparison with SliceGPT and ShortGPT on LLaMA-7B without fine-tuning
> >
> > We thank the reviewer for highlighting the gap between our data-free, fine-tuning-free method and other data-driven approaches on LLaMA-7B. In response, we have compared two newly released model compression methods, ShortGPT and SliceGPT, applied to LLaMA-7B without fine-tuning. The updated results are presented in the table below. Notably, Model Folding achieves comparable perplexity (PPL) to the data-driven pruning method ShortGPT on WikiText2. The updated details are provided in Table 1 of the revised PDF, with changes highlighted in yellow.
> >
> > | Prune ratio | Method           | Data usage  | WikiText2 ↓ | BoolQ | WinoGrande | ARC-e | ARC-c | Average ↑ |
> > | ----------- | ---------------- | ----------- | ----------- | ----- | ---------- | ----- | ----- | --------- |
> > | **0%**      | LLaMA-7B[1] | /           | 5.68        | 75.05 | 69.93      | 75.34 | 41.89 | 65.55     |
> > | **20%**     | Magnitude Prune  | /           | 36136       | 43.21 | 49.40      | 27.23 | 21.59 | 35.36     |
> > | **20%**     | LLM-Pruner[2]       | Gradients   | 10.53       | 59.39 | 61.33      | 59.18 | 37.18 | 54.27     |
> > | **20%**     | FLAP[3]             | Calibration | 6.87        | 69.63 | 68.35      | 69.91 | 39.25 | 61.79     |
> > | **20%**     | Wanda_sp[4]         | Calibration | 8.22        | 71.25 | 67.09      | 71.09 | 42.58 | 63.00     |
> > | **20%**     | SliceGPT[5]         | Calibration | 7.00        | 57.80 | 67.96      | 62.67 | 36.01 | 56.11     |
> > | **20%**     | ShortGPT[6]         | Calibration | 15.48       | 62.17 | 67.40      | 58.88 | 31.91 | 55.09     |
> > | **20%**     | Model Folding    | /           | 13.33       | 62.29 | 62.19      | 49.83 | 26.37 | 50.17     |
> >
> > [1] Touvron, Hugo, et al. "Llama: Open and efficient foundation language models." arXiv preprint arXiv:2302.13971 (2023).
> >
> > [2] Ma, Xinyin, Gongfan Fang, and Xinchao Wang. "LLM-pruner: on the structural pruning of large language models." Proceedings of the 37th International Conference on Neural Information Processing Systems. 2023.
> >
> > [3] An, Yongqi, et al. "Fluctuation-based Adaptive Structured Pruning for Large Language Models." arXiv preprint arXiv:2312.11983 (2023).
> >
> > [4] Sun, Mingjie, et al. "A simple and effective pruning approach for large language models." arXiv preprint arXiv:2306.11695 (2023).
> >
> > [5] Ashkboos, Saleh, et al. "Slicegpt: Compress large language models by deleting rows and columns." arXiv preprint arXiv:2401.15024 (2024).
> >
> > [6] Men, Xin, et al. "Shortgpt: Layers in large language models are more redundant than you expect." arXiv preprint arXiv:2403.03853 (2024).

---

> > > ### Comment · Reviewer_Ao8L · 2024-11-28
> > >
> > > Thank you for sharing the additional results. In my opinion, data-free compression might not be particularly critical for modern LLMs, as they are generally trained on extensive datasets, and some form of data is usually accessible. That said, if the paper aims to address specific scenarios, such as working with private or restricted data, it would be important to see more experiments demonstrating that the proposed data-free method offers clear advantages over approaches that leverage general, publicly available data while handling LLMs trained/finetuned on restricted data. So, I tend to keep my initial rating.

---

> > > > ### Author Response · Authors · 2024-11-30
> > > >
> > > > Thank you for raising this important question. While modern LLMs are trained on extensive datasets, access to such data or related domains is not always feasible in real-world scenarios. In regulated industries such as healthcare, finance, or defense, where data is often sensitive or proprietary, even general public datasets may not be suitable for fine-tuning or compression. Our work specifically addresses data-free settings, offering a robust solution for compressing LLMs without requiring any data or fine-tuning.
> > > >
> > > > To illustrate the importance of this setting, we demonstrate that using a suboptimally chosen, out-of-distribution (OOD) calibration dataset can result in worse performance compared to our data-free Model Folding approach. For example, we generated a dataset of random Hungarian words in repeated sequences and applied the Wanda compression method to LLaMA-7B. Although LLaMA-7B was trained on some Hungarian text, the language is underrepresented in its training corpus. Using this OOD calibration dataset, the perplexity on the WikiText2 benchmark increased from 8.22 (with the original C4 dataset) to 13.98. A similar performance drop (perplexity = 13.94) was observed with a Ukrainian dataset, highlighting the sensitivity of data-driven methods like Wanda to the domain alignment of the calibration data. These results highlight the robustness of data-free approaches like Model Folding in scenarios where appropriate calibration data is unavailable. Note that further optimization of these experiments is possible (we explored only a limited set of options), yet they showcase the challenges faced by data-driven methods with OOD calibration data.
> > > >
> > > > Although we cannot update the paper at this stage to include these findings, we sincerely thank the reviewer for raising this important point, which has further strengthened our motivation and work. We hope this response addresses your concerns, and if you find it satisfactory, we would greatly appreciate your support in revising your score. Thank you!

---

> ### Author Response · Authors · 2024-12-02
>
> Dear reviewer, today is the final day of the discussion period. We have submitted our rebuttal and would greatly appreciate any additional feedback you may have. Thank you for your time!

---

### Official Review · Reviewer_TqyN · 2024-11-03

**Soundness:** 3
**Presentation:** 3
**Contribution:** 3
**Rating:** 6
**Confidence:** 5

**Summary:**

The paper presents Model Folding, a data-free and fine-tuning-free model compression technique that merges structurally similar neurons within a neural network. This approach contrasts with traditional methods that often require access to training data or fine-tuning after compression.  Experiments on standard benchmarks, including ResNet and large models like LLaMA-7B, show that Model Folding achieves high sparsity while maintaining competitive performance with data-driven methods and outperforming recent data-free compression techniques.

**Strengths:**

1. The method has both theoretical justification and empirical support, demonstrating that k-means clustering is an optimal method for weight fusion in a data-free manner. Results from benchmarks like ResNet18 and LLaMA-7B show that model folding achieves performance on par with or surpasses existing data-driven and data-free compression methods, particularly at high sparsity levels.

2. Model folding is designed to be completely data-free, which differentiates it from other compression methods that typically require fine-tuning with original data. The authors propose two data-free alternatives for adjusting internal data statistics—Fold-AR and Fold-DIR—based on approximate and deep inversion-based techniques. It presents significant improvements over existing state-of-the-art data-free methods, demonstrating its applicability in compressing large-scale models effectively

**Weaknesses:**

1. More data-free and training-free references are needed, such as [1]

2. Lack of sufficient experimental results on compression ratio and speedup ratio.

[1] Haroush, Matan, et al. "The knowledge within: Methods for data-free model compression." Proceedings of the IEEE/CVF Conference on Computer Vision and Pattern Recognition. 2020.

**Questions:**

1. Is there a speed comparison of inference on different devices after compression of the model?

---

> ### Author Response · Authors · 2024-11-22
> **Rebuttal to Reviewer TqyN**
>
> We appreciate the reviewer’s thoughtful comments and questions on our work, as well as their emphasis on the importance of data-free approaches for adjusting batch normalization statistics.
>
>
> ## W.1: Additional references
>
> We have added additional references on data-free and fine-tuning-free methods, as suggested by the reviewer, highlighted in the updated PDF with a yellow background.
>
> ## W.2: Compression ratios and running time
>
> The table below shows the average runtime (over 10 runs on the CIFAR10 test dataset) and the sparsity of the folded ResNet18 model pre-trained on CIFAR10, evaluated on an NVIDIA A100 GPU. To ensure consistent measurements, we applied GPU synchronization and a warm-up phase.
>
> | Sparsity | 0.00   | 0.09   | 0.19   | 0.27   | 0.36   | 0.44   | 0.51   | 0.58   | 0.64   | 0.70   | 0.75   |
> |----------|--------|--------|--------|--------|--------|--------|--------|--------|--------|--------|--------|
> | avg runtime (seconds) | 0.647 | 0.683 | 0.717  | 0.642  | 0.722  | 0.554 | 0.615 | 0.588 | 0.534 | 0.534 | 0.416 |
>
>
> ## Q.1: Inference speed on edge devices
>
> We applied model folding to a LeNet5 model pre-trained on FashionMNIST with varying sparsity levels and evaluated the folded models on NVIDIA Jetson Nano, ESP-EYE, and Arduino Nano 33 BLE. All models were converted to float32 TensorFlow Lite format and executed uniformly across all devices. The updated details are also provided in Appendix L, highlighted with a yellow background.
>
> | Sparsity                  | 10%                           | 25%                           | 50%                          | 70%                          |
> | ------------------------- | ----------------------------- | ----------------------------- | ---------------------------- | ---------------------------- |
> | NVIDIA Jetson Nano[1]        | 2ms, RAM 59.5K, Flash 3.4M    | 2ms, RAM 55.7k, Flash 2.8M    | 1ms, RAM 48.0k,Flash 1.9M    | 1ms, RAM 36.5k,Flash 1.2M    |
> | ESP-EYE[2]                   | 2591ms,RAM 59.5K,Flash 3.4M   | 1868ms, RAM 55.7k, Flash 2.8M | 1532ms, RAM 48.0k,Flash 1.9M | 1186ms, RAM 36.5k,Flash 1.2M |
> | Arduino Nano 33 BLE Sense[3] | 6831ms, RAM 59.5k, Flash 3.4M | 3726ms, RAM 55.7k, Flash 2.8M | 4218ms, RAM 48.0k,Flash 1.9M | 2969ms, RAM 36.5k,Flash 1.2M |
>
> [1] NVIDIA. Jetson Nano - NVIDIA Developer. 2024. NVIDIA, https://www.nvidia.com/en-us/autonomous-machines/embedded-systems/jetson-nano/product-development/. Accessed 19 Nov. 2024.
>
> [2] Espressif Systems. ESP-EYE Development Board - Espressif Systems. 2024. Espressif Systems, https://www.espressif.com/en/products/devkits/esp-eye/overview. Accessed 19 Nov. 2024.
>
> [3] Arduino. Arduino Nano 33 BLE Documentation. 2024. Arduino, https://docs.arduino.cc/hardware/nano-33-ble/. Accessed 19 Nov. 2024.

---

> > ### Comment · Reviewer_TqyN · 2024-11-26
> >
> > I appreciate the author's additional experiments. But similar to reviewers Ao8L and AJd2, considering that the proposed method in this paper lack sufficiently impressive innovation, I decided to maintain my score.

---

> > > ### Author Response · Authors · 2024-11-27
> > >
> > > Thank you for your feedback and for taking the time to review our work. We would like to address your concerns regarding the innovation of our proposed method.
> > >
> > > We view the novelty of Model Folding in its ability to successfully tackle a problem previously addressed by recent works while offering significant advancements through the following contributions:
> > >
> > > - **Theoretical grounding**: We provide a mathematical foundation for Model Folding by leveraging a factorization-based formulation of k-Means clustering. This formulation demonstrates the optimality of Model Folding for compressing model weights across various layer types, as detailed in the main paper and the appendix. Furthermore, it establishes a novel connection between k-Means clustering and the Linear Assignment Problem (LAP), unifying the model compression and model merging settings. This theoretical insight is thoroughly discussed in our response to Reviewer Ao8L (we sincerely appreciate the reviewer for raising this question) and elaborated in the newly added Appendix C.
> > >
> > >
> > >
> > > - **Innovative data-free approximations**: Building on this theory, we propose two novel approaches (Fold-AR and Fold-DIR) that provide data-free approximations of REPAIR. These methods are particularly effective, achieving superior performance compared to state-of-the-art data-driven methods such as IFM and INN. For example, at 75% sparsity on ResNet18 pre-trained on CIFAR10, while IFM suffers a significant performance drop to ~10% accuracy, Model Folding maintains ~84% accuracy, demonstrating its robustness at high sparsity levels.
> > >
> > >
> > >
> > > - **Uniqueness in the data-free setting for LLM compression**: Model Folding stands out as the only method, to our knowledge, capable of achieving performance close to SOTA LLM compression methods without requiring any data or fine-tuning. This makes it a highly valuable solution for applications where access to sensitive or proprietary data is restricted due to privacy or security concerns.
> > >
> > > We believe these contributions represent both theoretical and practical innovations, advancing the field of data-free model compression and addressing a critical gap in existing methods. We hope this clarifies the novelty and significance of our work.
> > >
> > > Thank you again for your detailed review and for considering our response.

---

### Official Review · Reviewer_mvsb · 2024-11-03

**Soundness:** 2
**Presentation:** 2
**Contribution:** 3
**Rating:** 6
**Confidence:** 2

**Summary:**

The paper proposes a new training- and fine-tuning-free method for neural network compression. Inspired by recent research on neuron alignment, in particular the weight matching algorithm [Ainsworth et al., 2023], the method essentially works by applying k-means vector quantization to cluster similar neurons (i.e., rows or channels of a weight matrix/tensor) together, and applying an additional data-free REPAIR (Jordan et al. (2022)) procedure to preserve the activation statistics of the original network. Experiments on convnets (e.g., ResNets, VGGs) demonstrate improved accuracy / sparsity tradeoff against other SOTA data-free pruning methods (e.g., FM (Chen et al., 2023), and INN (Solodskikh et al., 2023)). Experiment on an LLM (LLaMA-7B) shows comparable / somewhat worse performance than other pruning methods that require data/finetuning.

**Strengths:**

I think conceptually connecting the research and recent methods in neuron alignment / neural network symmetry (Yamada et al., 2023; Ainsworth et al., 2023) to the problem of model compression is somewhat novel and deserves more attention, although it certainly has been done many times, see e.g., [Zhou et al., 2018](https://arxiv.org/abs/1804.05862) and the paper [Chen et al., 2023](https://arxiv.org/pdf/2310.06756) cited in the current work. Methodologically the contribution seems fairly incremental (i.e., weight quantization, extension of the existing REPAIR (Jordan et al. (2022)) method).

The writing quality and clarity is mostly good; the experiments/figures are informative.

I rate the significance as moderate. In terms of practical impact, the method is relatively easy to understand and implement, and obtains competitive performance among other data- and tuning-free approaches. However, I was hoping to see more in terms of bridging the conceptual understanding of neural network symmetry and compression, e.g., whether the proposed *model folding* method (essentially VQ) compares/relates to the weight matching of (Ainsworth et al., 2023), and whether it also enables some degree of alignment or mode connectivity b/w two randomly pretrained neural networks.

**Weaknesses:**

1. The paper is missing related literature and baselines on model quantization. A simple Google search suggests quite a few related papers, e.g., using vector quantization for model compression [[Martinez et al., 2021]](https://openaccess.thecvf.com/content/CVPR2021/papers/Martinez_Permute_Quantize_and_Fine-Tune_Efficient_Compression_of_Neural_Networks_CVPR_2021_paper.pdf) and post-training quantization [[Nagel et al., 2021](https://arxiv.org/pdf/2106.08295)] which is also training-free.

2. Some of the writing can be improved:
  - The section on Fold-AR is a bit hard to understand; it's unclear what exactly is being proposed. Perhaps an algorithm box can help.
  - The paper repeatedly talks about "merging similar channels" as the basis of the model folding algorithm, without defining it for various network architectures considered (especially outside the context of CNNs). This should be clarified, as how the scalar weights are grouped into vectors can have a big performance impact on VQ.
  - Some typographical errors: Line 278: "The variance ratio of the l-the", "l-the" -> "l-th"; unfinished sentence on Line 414: "Fold-DIR ourperforms Fold-AR as the cost of generating a batch of synthetic images and a forward pass through the network."

**Questions:**

As mentioned earlier, I find the connection between weight matching (Ainsworth et al., 2023) and the proposed VQ method to be somewhat lacking, and the former seems more like an inspiration on which the method is loosely based. Do the authors suspect a similar effect of alignment to be achieved through quantization, such that distant modes can be connected with low loss barrier in between them (as in (Ainsworth et al., 2023))?

---

> ### Author Response · Authors · 2024-11-22
> **Rebuttal to Reviewer mvsb  (1/5)**
>
> Thank you for your thoughtful feedback, particularly for asking to highlight the connection between model folding and weight matching. We tackle this question first before addressing the weaknesses listed in the review.
>
>
>
> ## Relationship between Weight Matching and Model Folding
>
> Weight Matching \[[Ainsworth, 2023](https://arxiv.org/abs/2209.04836)\] fuses two models into one, whereas Model Folding compresses the weight tensors/matrices of a single network. While inspired by weight matching, Model Folding addresses a distinct use case, leading to different optimization problems (K-Means vs. LAP). Notably, the Linear Sum Assignment Problem (LAP) can be framed as a constrained K-Means variant, where each cluster contains exactly two vectors - one from network A and one from network B.
>
> As an example for this discussion, consider a simple feedforward network. The steps of our proposed compression algorithm involve iteratively solving the following:
>
> $$
> \mathbf{C}_l = \mathop{\mathrm{argmin}}\_{\mathbf{C}_l} \lVert\mathbf{W}_l - \mathbf{C}_l \mathbf{W}_l\lVert_F^2 + \lVert{\mathbf{W}}\_{l+1}^T - \mathbf{C}_l {\mathbf{W}}\_{l+1}^T\lVert_F^2,
> $$
>
> such that $\mathbf{C}_l = \mathbf{U}_l(\mathbf{U}_l^T\mathbf{U}_l)\mathbf{U}_l^T$ where $\mathbf{U}_l^T$ is a clustering matrix.
>
> Weight Matching merges two feedforward networks by iteratively optimizing:
>
> $$
> \mathbf{P}_l = \mathop{\mathrm{argmin}}\_{\mathbf{P}_a} \lVert{\mathbf{W}}\_{A,l} - \mathbf{P}_l {\mathbf{W}}\_{B,l}\lVert_F^2 + \lVert{\mathbf{W}}\_{A, l+1}^T - \mathbf{P}_l {\mathbf{W}}\_{B, l+1}^T\lVert_F^2,
> $$
>
> where $\mathbf{P}_l$ is a permutation matrix. To connect Weight Matching with our method, we frame our approach within the model merging domain. This begins by establishing a relationship between K-Means and the Linear Sum Assignment (LAP) problem.
>
> Given two matrices, $\mathbf{W}_A$ and $\mathbf{W}_B$, where the rows are treated as vectors to be combined using unconstrained K-Means, the problem can be formulated as follows:
>
>
> $$
> \\begin{align}
> \\mathbf{C} = \\mathop{\\mathrm{argmin}}\_{\\mathbf{C}} \\left\\|\\begin{bmatrix}\\mathbf{W}_A \\\\ \\mathbf{W}_B\\end{bmatrix} - \\mathbf{C}\\begin{bmatrix}\\mathbf{W}_A \\\\ \\mathbf{W}_B\\end{bmatrix}\\right\\|_F^2
> \end{align}
> $$
>
> such that $\mathbf{C} = \mathbf{U}(\mathbf{U}^T\mathbf{U})\mathbf{U}^T$ where $\mathbf{U}^T$ is a clustering matrix.
>
> If we constrain $\mathbf{C}$ such that $\mathbf{C} = \begin{bmatrix} \mathbf{P} & \mathbf{I} \end{bmatrix}$, where $\mathbf{P}$ is a permutation matrix, the following observations can be made:
>
> 1. The matrix $\mathbf{C} = \begin{bmatrix} \mathbf{P} & \mathbf{I}\end{bmatrix}$ is still a clustering matrix.
> 2. The matrix $\mathbf{C}$ obtaines the form:
>
> $$
> \\mathbf{C} = \\frac{1}{2}\\begin{bmatrix}\\mathbf{P} \\\\\\mathbf{I}\\end{bmatrix} \\begin{bmatrix}\\mathbf{P}^T & \\mathbf{I}\\end{bmatrix} = \\frac{1}{2}\begin{bmatrix}
>   	\\mathbf{I} & \\mathbf{P} \\\\
>   	\\mathbf{P}^T & \\mathbf{I}\\end{bmatrix},
> $$
>
> reducing the problem defined by Eq. $(1)$ to:
>
> $$
> \\mathbf{P} = \\mathop{\\mathrm{argmin}}\_{\\mathbf{P}} \\|\\begin{bmatrix}\\mathbf{W}_A \\\\ \\mathbf{W}_B\\end{bmatrix} - \\mathbf{P} \\begin{bmatrix}\\mathbf{W}_A \\\\ \\mathbf{W}_B\\end{bmatrix}\\|_F^2,
> $$
>
> which is an instance of the Linear Sum Assignment Problem, as minimizing the cost:
> $$
> J = \\|\\begin{bmatrix}\\mathbf{W}_A \\\\ \\mathbf{W}_B\\end{bmatrix} - \\mathbf{P} \\begin{bmatrix}\\mathbf{W}_A \\\\ \\mathbf{W}_B\\end{bmatrix}\\|_F^2
> $$
>
>
> is equivalent to maximizing:
>
> $$
> J^+ = \\text{tr}\\left(\\mathbf{P}\\begin{bmatrix}\\mathbf{W}_A \\\\ \\mathbf{W}_B\\end{bmatrix}\\begin{bmatrix}\\mathbf{W}_A \\\\ \\mathbf{W}_B\\end{bmatrix}^T \\right)
> $$
>
>
> Building on the previous results, we define Model Folding for merging networks as follows:
>
> $$
> J_l = \\left\\|\\begin{bmatrix}
>     	\\mathbf{W}\_{l,A} \\\\
>     	\\mathbf{W}\_{l,B}
> 	\\end{bmatrix} - \\mathbf{C}_l \\begin{bmatrix}
>     	\\mathbf{W}\_{l,A} \\\\
>     	\\mathbf{W}\_{l,B}
> 	\\end{bmatrix}\\right\\|_F^2 + \\left\\|\\begin{bmatrix}
>     	\\mathbf{W}\_{l+1,A} &
>     	\\mathbf{W}\_{l+1,B}
> 	\\end{bmatrix} - \\begin{bmatrix}
>     	\\mathbf{W}\_{l+1,A} &
>     	\\mathbf{W}\_{l+1,B}
> 	\\end{bmatrix}\\mathbf{C}\_{l}^T\\right\\|_F^2.
> $$
>
>
> and constraining $\\mathbf{C}_l$ to $\\mathbf{C} = \\begin{bmatrix} \\mathbf{P} & \\mathbf{I}\\end{bmatrix}$, where $\\mathbf{P}$ is a permutation matrix. With this we indeed get the Weight Matching \[[Ainsworth, 2023](https://arxiv.org/abs/2209.04836)\] coordinate descent cost:
> $$
> 	J_l = \\frac{1}{2} \\left\\|\\mathbf{W}\_{l,A} - \\mathbf{P}_l \\mathbf{W}\_{l,B} \\right\\| + \\frac{1}{2} \\left\\|\\mathbf{W}\_{l+1,A}^T - \\mathbf{P}_l \\mathbf{W}\_{l+1,B}^T \\right\\|.
> $$
>
>
> We have also emphasized this in Table 2 and Table 3, located in Appendix C of the updated PDF.

---

> ### Author Response · Authors · 2024-11-22
> **Rebuttal to Reviewer mvsb (2/5)**
>
> ## Model Folding for connecting models
>
> We provide a small experimental setup comparing **WM** \[[Ainsworth, 2023](https://arxiv.org/abs/2209.04836)\], **ZipIt!** \[[Stoica, 2023](https://arxiv.org/abs/2305.03053)\] and our method for merging networks trained on the same task and networks trained on separate tasks.
>
> For the experiments involving the merging of networks trained on disjoint tasks, we used instances of VGG11 and ResNet18 trained on CIFAR10 with a 5+5 label split. All experiments were performed with REPAIR.
>
> ##### Merging networks trained on separate tasks
> | Model      	| WM     	| ZipIt! 	| Ours   	|
> |----------------|------------|------------|------------|
> | VGG11      	| 0.57   	| 0.69   	| **0.71**   |
> | ResNet18      | 0.48     	| 0.74     	| **0.75**  |
>
> For the experiments involving the merging of networks trained on the same task, we used instances of VGG11 and ResNet18, both trained on CIFAR10. All experiments were performed with REPAIR.
>
>
> ##### Merging networks trained on the same task
> | Model      	| WM     	| ZipIt! 	| Ours   	|
> |----------------|------------|------------|------------|
> | VGG11      	| 0.89   	| 0.87   	| **0.92**    |
> | ResNet18      | 0.92    	| 0.91     	| **0.93**    |
>
> We also highlighted in appendix C of the updated pdf.
>
>
> ## W.1: Related literature and baselines on model quantization
>
> We acknowledge the relevance of the first paper [[Martinez et al., 2021\]](https://openaccess.thecvf.com/content/CVPR2021/papers/Martinez_Permute_Quantize_and_Fine-Tune_Efficient_Compression_of_Neural_Networks_CVPR_2021_paper.pdf), which leverages redundancies among parameter groups by computing centroids over parameter subvectors to create a codebook for each layer. While this is conceptually related, their approach requires fine-tuning as a post-quantization step and is therefore not data-free. Additionally, clustering at the subvector level, as proposed in [[Martinez et al., 2021\]](https://openaccess.thecvf.com/content/CVPR2021/papers/Martinez_Permute_Quantize_and_Fine-Tune_Efficient_Compression_of_Neural_Networks_CVPR_2021_paper.pdf), differs substantially from clustering at the level of hidden units, which is the focus of all weight-matching literature. We observe that the cited method, by allowing one axis (columns) of the weight matrix $\mathbf{W}$ to be partitioned into subvectors, results in a quantization that is not structured. Specifically, during inference, the compressed codebook must be cast into a matrix $\hat{\mathbf{W}}$ with the same dimensions as $\mathbf{W}$, but with **entries shared** from the codebook. This indicates that the resulting network compression lacks structure, analogous to the distinction between structured and unstructured pruning. In contrast, our method enforces structured compression by allowing only one group per weight axis. As demonstrated by **Lemma 1** (Appendix B), it achieves this by combining the output channels of a layer, aligning with the term "folding" in the naming of our method. This unstructured nature of [[Martinez et al., 2021\]](https://openaccess.thecvf.com/content/CVPR2021/papers/Martinez_Permute_Quantize_and_Fine-Tune_Efficient_Compression_of_Neural_Networks_CVPR_2021_paper.pdf) is further validated by examining **Line 41** in ([CompressedLinear.py](https://github.com/uber-research/permute-quantize-finetune/blob/main/src/compressed_layers/CompressedLinear.py)), where, during the forward pass, the codes are reorganized into a matrix matching the shape of the original uncompressed network. In our paper we only compare structured compression methods. Exploring a combination of subvector-level clustering with a variant of data-free REPAIR is an interesting direction for future research.
>
> The second paper [[Nagel et al., 2021\]](https://arxiv.org/pdf/2106.08295) focuses on scalar quantization and does not modify the network's width. This method is orthogonal to model folding, as folded models (similar to pruned models) can also undergo quantization. While we cite a few quantization-related works in our submission, we view this and similar approaches as complementary rather than competitive to model folding.

---

> ### Author Response · Authors · 2024-11-22
> **Rebuttal to Reviewer mvsb (3/5)**
>
> ## W.2a: Fold-AR is hard to follow
>
> We have included detailed derivations and an algorithmic box in Appendix G. All changes to the paper are highlighted.
>
>
>
> ## W.2b: Folding similar channels in LlamaMLP
>
> The LlamaMLP module is composed of three sub-layers: `gate_proj`, `up_proj`, and `down_proj`. These sub-layers define the structure and functionality of the MLP, with the main computation pipeline expressed as `down_proj(act_fn(gate_proj(x))×up_proj(x))`. We cluster similar channels in both the output channel and input channel of each sub layer.
>
> To fold the **input channels** of LlamaMLP, we simultaneously consider the input dimensions of both `gate_proj` and `up_proj` layers, as they collectively define the effective input to the `gate_up` sub-layer,and just clustering the input channels of `gate_proj` and `up_proj`respectively according to methods we applied in `MLP layers`.
>
> To fold the **output channels** of LlamaMLP, we firstly consider the output channels of both `gate_proj` and `up_proj` layers by clustering and adjusting the input channel of the `down_proj`. We then adjust the output channel of `down_proj` according to the residual connection used out of LlamaMLP.
>
> We also highlighted in appendix J of the updated pdf.
>
>
>
> ## W.2b: Folding similar channels in LlamaAttention
>
> The LlamaAttention module consists of four primary sub-layers: `q_proj`, `k_proj`, `v_proj`, and `o_proj`. These sub-layers define the query, key, value, and output projections respectively. For clarity and simplicity, we conceptualize `q_proj`, `k_proj`, and `v_proj` as a unified sub-layer, referred to as `q_k_v`, which computes the intermediate representations required for attention calculations. The `o_proj` sub-layer processes the final output of the attention mechanism. We treat the attention head as the structure to be folded in LlamaAttention. By reshaping the weights of each sub-layer into an MLP-like tensor, we can cluster similar heads, similar to how it is done for a standard MLP layer. The following sections provide the implementation details.
>
>
>
> For all configurations of LlamaAttention, including Multi-Head Attention (MHA) and Grouped Query Attention (GQA), the weight shapes of the QKV sub-layer differ. In MHA, the weights for q, k, and v projections share the same shape, [num_heads×head_dim,hidden_size]. In GQA, however, the weights for k and v projections have the shape [num_kv_heads×head_dim,hidden_size].
>
>
>
> When performing **output channel folding** for the LlamaAttention layer, the clustering of the `o_proj` sub-layer’s output channels is dictated by the residual connection out of LlamaAttention, ensuring alignment with the clustering results from previous modules. Specifically, the `o_proj` weights, originally shaped as [num_heads×head_dim,hidden_size], are first reshaped into [num_heads,head_dim,hidden_size], clustered along the first dimension (num_heads), and then reshaped back to their original form. To perform output channel clustering within the QKV sub-layer, the weights of q, k, and v are first reshaped into [num_heads,head_dim,hidden_size] (or [num_kv_heads,head_dim,hidden_size] for k and v in GQA) and clustered along the first dimension (num_heads or num_kv_heads). After clustering, the weights are reshaped back to their original dimensions.
>
> To fold the **input channel folding**, the focus is on the input channels of q, k, and v weights. Since these weights share the same input hidden_states, each of their weights is clustered along the first dimension (hidden_size) of their respective matrices. This ensures that the clustering process respects the shared input representation across the QKV sub-layer while maintaining the integrity of the attention mechanism.
>
> We also highlighted in appendix J of the updated pdf.

---

> ### Author Response · Authors · 2024-11-22
> **Rebuttal to Reviewer mvsb (4/5)**
>
> ## W.2b: Folding similar channels in MLP
>
>
> For fully connected networks, where two successive layers are defined as:
> $$
> \\mathbf{x}_{l} = \\sigma(\\mathbf{W}\_{l}\\mathbf{x}\_{l-1}) \\;\\; \\text{and} \\;\\; \\mathbf{x}\_{l+1} = \\sigma(\\mathbf{W}\_{l+1}\\mathbf{x}_l)
> $$
> where $\\mathbf{x}l$ represents the activations of layer $l$, $\\mathbf{W}l$ and $\\mathbf{W}{l+1}$ are the weight matrices, and $\\sigma$ is the activation function. The channels of the layer are defined as the coordinates $\mathbf{x}{l,i}$ of the vector $\\mathbf{x}_l$. Each channel corresponds to a specific dimension in the activations.
>
>
> The folding cost $J_l$ for the $l$-th layer is defined as:
> $$
> 	J_l = \\left\\| \\mathbf{W}_l - \\mathbf{C}_l \\mathbf{W}_l\\right\\|_F^2 + \\left\\| \\mathbf{W}\_{l+1}^T - \\mathbf{C}_l \\mathbf{W}\_{l+1}^T\\right\\|_F^2
> $$
> where $\\mathbf{C}_l$ is a clustering matrix. This cost function represents the optimization objective to minimize the approximation error introduced by folding (clustering) the weights of the $l$-th layer. The first term measures the reconstruction error for the weights $\\mathbf{W}l$, while the second term measures the reconstruction error for the weights $\\mathbf{W}{l+1}$ under the transformation $\\mathbf{C}_l$. Together, these terms ensure that the clustering transformation preserves the structure and relationships of the weights across layers.
>
>
>
> From the perspective of K-Means as a matrix decomposition problem, the grouping of scalar weights into vectors is defined as follows:
>
> $$
> \\mathbf{W}_l = \\begin{bmatrix}
>     	\\mathbf{p}_1^T \\\\
>     	\\mathbf{p}_2^T \\\\
>     	\\vdots \\\\
>     	\\mathbf{p}_n^T
> 	\\end{bmatrix} \\;\\; \\text{and} \\;\\; \\mathbf{W}\_{l+1} = \\begin{bmatrix}
>     	\\mathbf{q}_1 &
>     	\\mathbf{q}_2 &
>     	\\ldots &
>     	\\mathbf{q}_n
> 	\\end{bmatrix}
> $$
>
> where $\mathbf{p}_i^T$ are the rows of $\mathbf{W}_l$ and $\mathbf{q}i$ are the columns of $\mathbf{W}{l+1}$. These groupings reflect the natural structure of the weight matrices in fully connected layers:
>
> Each row of $\mathbf{W}_l$ represents the weights associated with a specific output channel of layer $l$.
>
> Each column of $\\mathbf{W}\_{l+1}$ represents the weights associated with a specific input channel of layer $l+1$.
>
> In this formulation, the rows $\mathbf{p}_i^T$ and columns $\mathbf{q}_i$ are treated as vectors to be clustered by the matrix $\mathbf{C}_l$, which aligns with the K-Means decomposition perspective. The clustering matrix $\mathbf{C}_l$ maps these weights into representative clusters, preserving the relationships between input and output channels across layers while enabling efficient compression.
>
> We also highlighted in appendix H of the updated pdf.

---

> > ### Author Response · Authors · 2024-11-22
> > **Rebuttal to Reviewer mvsb (5/5)**
> >
> > ## W.2b: Folding similar channels in convolutional layers
> >
> >
> > In case of convolutional layers, whose two successive layers are defined as:
> > $$
> > \\begin{align}
> > 	\\mathcal{X}\_{l} = \\sigma(\\mathcal{W}\_{l} * \\mathcal{X}\_{l-1}) \\;\\; \\text{and} \\;\\; \\mathcal{X}\_{l+1} = \\sigma(\\mathcal{W}\_{l+1}\\mathcal{X}_l)
> > \\end{align}
> > $$
> >
> > where $\\mathcal{X}l$ is a 3-dimensional feature tensor with values $\\mathcal{X}^{(l)}{c_o,i,j}$. The first dimension, $c_o$, corresponds to the output channels, while $i$ and $j$ correspond to spatial pixel locations. The 4-dimensional weight tensor $\\mathcal{W}l$ has values $\mathcal{W}^{(l)}{c_o, c_i, i,j}$, where:
> > $c_o$ corresponds to the output channels of $\\mathcal{X}_l$.
> > $c_i$ corresponds to the input channels of $\\mathcal{X}\_{l-1}$.
> > We decompose the weight tensor $\mathcal{W}l$ such that output channels of $\mathcal{X}l$ (values $\\mathcal{X}^{(l)}{c_o,i,j}$ for $c_o = 1, \ldots, c{out}$), which are similar in some sense, get merged. The folding problem is defined as:
> > $$
> > \\begin{align*}
> > 	J_l = \\left\\|\\mathcal{W}_l - \\mathcal{C}_l \\circ \\mathcal{W}_l\\right\\|_T^2 + \\left\\|\\mathcal{W}\_{l+1} - \\mathcal{W}\_{l+1} \\circ \\mathcal{C}_l\\right\\|_T^2
> > \\end{align*}
> > $$
> >
> > where $\\mathcal{C}l$ corresponds to a $1 \\times 1$ convolution parameterized by the clustering matrix $\\mathbf{C}l$ as $\\mathcal{C}^{(l)}{c, 1, 1} = \\mathbf{C}{l, c, c'}$.
> >
> > From this definition it follows that:
> > $$
> > \\begin{align*}
> > 	J_l = \\left\\|\\mathbf{W}_l - \\mathbf{C}_l \\mathbf{W}_l\\right\\|_T^2 + \\left\\|\\mathbf{W}\_{l+1} - \\mathbf{W}\_{l+1}\\mathbf{C}_l^T\\right\\|_T^2
> > \\end{align*}
> > $$
> >
> > The weight tensors $\\mathcal{W}l$ and $\\mathcal{W}{l+1}$ are mapped to matrices $\\mathbf{W}l$ and $\\mathbf{W}{l+1}$ as:
> > $$
> > \\begin{align*}
> > 	\\mathbf{W}_l = \\begin{bmatrix}
> >     	\\text{vec}(\\mathcal{W}^{(l)}\_{1, 1, :, :})^T & \\text{vec}(\\mathcal{W}^{(l)}\_{1, 2, :, :})^T & ... & \\text{vec}(\\mathcal{W}^{(l)}\_{1, c\_{in}, :, :})^T \\\\
> >     	\\text{vec}(\\mathcal{W}^{(l)}\_{2, 1, :, :})^T & \\text{vec}(\\mathcal{W}^{(l)}\_{2, 2, :, :})^T & ... & \\text{vec}(\\mathcal{W}^{(l)}\_{2, c\_{in}, :, :})^T \\\\
> >     	\\vdots  & \\vdots & ... & \\vdots \\\\
> >     	\\text{vec}(\\mathcal{W}^{(l)}\_{c\_{out}, 1, :, :})^T & \\text{vec}(\\mathcal{W}^{(l)}\_{c\_{out}, 2, :, :})^T & ... & \\text{vec}(\\mathcal{W}^{(l)}\_{c\_{out}, c\_{in}, :, :})^T \\\\
> > 	\\end{bmatrix}.
> > \\end{align*}
> > $$
> >
> > This means that each convolutional filter contributing to an output channel $c_o$ is flattened and stacked into a vector, forming the $c_o$-th row of the matrix $\\mathbf{W}l$. Similarly, for $\\mathcal{W}{l+1}$, each filter associated with the $c_i$-th input channel is flattened and stacked into a vector, forming a column of the matrix $\mathbf{W}_{l+1}$:
> > $$
> > \\begin{align*}
> > 	\\mathbf{W}\_{l+1} = \\begin{bmatrix}
> >     	\\text{vec}(\\mathcal{W}^{(l+1)}\_{1, 1, :, :}) & \\text{vec}(\\mathcal{W}^{(l+1)}\_{1, 2, :, :}) & ... & \\text{vec}(\\mathcal{W}^{(l+1)}\_{1, c\_{in}, :, :}) \\\\
> >     	\\text{vec}(\\mathcal{W}^{(l+1)}\_{2, 1, :, :}) & \\text{vec}(\\mathcal{W}^{(l+1)}\_{2, 2, :, :}) & ... & \\text{vec}(\\mathcal{W}^{(l+1)}\_{2, c\_{in}, :, :}) \\\\
> >     	\\vdots  & \\vdots & ... & \\vdots \\\\
> >     	\\text{vec}(\\mathcal{W}^{(l+1)}\_{c\_{out}, 1, :, :}) & \\text{vec}(\\mathcal{W}^{(l+1)}\_{c\_{out}, 2, :, :}) & ... & \\text{vec}(\\mathcal{W}^{(l+1)}\_{c\_{out}, c\_{in}, :, :}) \\\\
> > 	\\end{bmatrix}.
> > \\end{align*}
> > $$
> >
> > where columns correspond to input channels. The rows $\\mathbf{p}_i^T$ of $\\mathbf{W}_l$ and columns $\\mathbf{q}j$ of $\\mathbf{W}{l+1}$ are grouped into clusters for the folding process, consistent with K-Means.
> >
> > From the perspective of K-Means as a matrix decomposition problem, the grouping of scalar weights into vectors is defined as follows:
> > $$
> > \\mathbf{W}_l = \\begin{bmatrix}
> >     	\\mathbf{p}_1^T \\\\
> >     	\\mathbf{p}_2^T \\\\
> >     	\\vdots \\\\
> >     	\\mathbf{p}_n^T
> > 	\\end{bmatrix} \\;\\; \\text{and} \\;\\; \\mathbf{W}\_{l+1} = \\begin{bmatrix}
> >     	\\mathbf{q}_1 &
> >     	\\mathbf{q}_2 &
> >     	... &
> >     	\\mathbf{q}_n
> > 	\\end{bmatrix}
> > $$
> >
> > where:
> >
> > $$\\mathbf{p}_i^T = \\begin{bmatrix}\\text{vec}(\\mathcal{W}^{(l)}\_{i, 1, :, :})^T & \\text{vec}(\\mathcal{W}^{(l)}\_{i, 2, :, :})^T & ... & \\text{vec}(\\mathcal{W}^{(l)}\_{i, c\_{in}, :, :})^T \\end{bmatrix}$$
> >
> > and:
> >
> > $$\\mathbf{q}_j = \\begin{bmatrix}
> >     \\text{vec}(\\mathcal{W}^{(l+1)}\_{1, j, :, :})^T &
> >     \\text{vec}(\\mathcal{W}^{(l+1)}\_{2, j, :, :})^T &
> >     ... &
> >     \\text{vec}(\\mathcal{W}^{(l+1)}\_{c\_{out}, j, :, :})^T
> > \\end{bmatrix}^T
> > $$.
> >
> >
> >
> > We also highlighted in appendix I of the updated pdf.
> >
> > ## W.2c: Typos
> >
> > The typos have been corrected. Thank you for your keen attention to detail!

---

> ### Author Response · Authors · 2024-11-28
>
> Thank you once again for your valuable feedback. As the end of the discussion period is approaching, we are hoping to hear your thoughts on our response. We hope that we have addressed your comments, and we would greatly appreciate it if you were willing to consider increasing your score if you are satisfied with our response.

---

> ### Author Response · Authors · 2024-12-02
>
> Dear reviewer, today is the final day of the discussion period. We have submitted our rebuttal and would greatly appreciate any additional feedback you may have. Thank you for your time!

---

> > ### Comment · Reviewer_mvsb · 2024-12-03
> >
> > Thank you for your response, which has addressed most of my concerns; I therefore increased my score slightly.
> > However I disagree with the argument that [Nagel et al., 2021] or similar methods based on scalar quantization (SQ) are irrelevant as baselines, as SQ can be thought of as a special case of VQ (with block size=1) and both use quantization to optimize for the model size v.s. performance trade-off.

---

> > > ### Author Response · Authors · 2024-12-03
> > >
> > > Thank you for your thoughtful feedback and for raising the score in support of our paper. We deeply value your time and consideration, and we agree that it would be intriguing to explore a comparison, including how model folding aligns with VQ methods, as part of our future work.

---

### Meta-Review · Area_Chair_bUKK · 2024-12-17

**Metareview:**

This paper presents a data-free method for merging columns in the weight matrix, consisting of two key steps. First, a clustering-based approach is used to identify and merge similar columns in the weight matrix. Second, observing that compressing the model can lead to variance ratio collapse or overshooting, the authors introduce data-free Fold-AR and Fold-DIR techniques to invert and repair the model. Experimental results demonstrate that the proposed method outperforms previous data-free approaches, particularly at high sparsity ratios.

Most reviewers find the paper well-written, the evaluation is comprehensive with theoretical justifications.

**Additional Comments On Reviewer Discussion:**

During the rebuttal, the authors clarified most concerns with extra theoretical justifications and experimental evaluations.

---

### Decision · Program_Chairs · 2025-01-22

Accept (Poster)